# Direct Selective Oxidation of Hydrogen Sulfide: Laboratory, Pilot and Industrial Tests

**Sergei Khairulin [1,\*], Mikhail Kerzhentsev [1], Anton Salnikov [1] and Zinfer R. Ismagilov [1,2]**

[1] FRC, Boreskov Institute of Catalysis SB RAS, 630090 Novosibirsk, Russia; ma_k@catalysis.ru (M.K.); salnikov@catalysis.ru (A.S.); zri@catalysis.ru (Z.R.I.)

[2] FRC of Coal and Coal Chemistry, Institute of Coal Chemistry and Chemical Materials Science SB RAS, 630090 Novosibirsk, Russia

\* Correspondence: sergk@catalysis.ru; Tel.: +73-833-306219

**Abstract:** This article is devoted to scientific and technical aspects of the direct catalytic oxidation of hydrogen sulfide for the production of elemental sulfur. It includes a detailed description of the Claus process as the main reference technology for hydrogen sulfide processing methods. An overview of modern catalytic systems for direct catalytic oxidation technology and known processes is presented. Descriptions of the scientific results of the Institute of Catalysis of the SB RAS in a study of the physical and chemical foundations of the process and the creation of a catalyst for it are included. The Boreskov Institute of Catalysis SB RAS technologies based on fundamental studies and their pilot and industrial testing results are described.

**Keywords:** gas purification; hydrogen sulfide; direct catalytic oxidation; fluidized catalyst bed; hydrogen sulfide removal facilities

## 1. Introduction

According to modern classification, hydrogen sulfide is a highly hazardous substance which contributes significantly to the pollution of the atmosphere. The destruction of vegetation, the death of aqueous flora and fauna, an increase in the incidences of cancer and diseases of the respiratory tract, and "acid" rain are typical direct consequences of the release of hydrogen sulfide and sulfur dioxide into the atmosphere. The main sources of hydrogen sulfide emissions into the atmosphere and water include mining and the processing of sulfurous natural gas and oil, coal gasification, and biomass processing [1–4].

In fact, 40% of global gas reserves currently identified as viable, i.e., more than 70 trillion nm$^3$, are "acidic", and more than 10 trillion Nm$^3$ contain more than 10% $H_2S$ [5].

To date, the estimated overall flow rate of the produced and processed sulfuric gas is about 100 billion m$^3$/year, and its contribution to the global mining of natural gases is 10–15%. At the same time, up to 60% of global sulfur production depends on the $H_2S$ in these sulfuric gases, and there is a steady increasing trend in the share of sulfur obtained in this manner in the global balance of elementary sulfur production [6].

Another typical example characterizing the overall situation is the disposal of sulfurous oil-associated gases formed during the extraction of sulfur oil. The total flow rate of deposits located in the densely populated areas of the Volga-Ural oil and gas province is up to 1 billion m$^3$/year. The involvement of such gases in the fuel and energy balance will save up to 1 million tons/year of fuel. However, the high hydrogen sulfide content (1–6%) precludes their use as hydrocarbon fuel supplied to the population, industrial enterprises, and as raw materials for the synthesis of chemical products.

At present, the torching of such gases leads to the contamination of the atmosphere with toxic sulfur di- and tri- oxide, sulfuric acid, products of incomplete burning of

hydrocarbons, and carcinogenic soot in amounts of up to one million tons per year. The average fraction of the incinerated associated oil gas in Russia was 24.4% in 2013 [7].

The ecological effects of burning are significantly worsened due to the flare disposal of hydrogen sulfide-containing oil-associated gases (OAG). The burning of one billion nm$^3$ of OAG results in atmospheric emissions of up to 60,000 tons of highly toxic $H_2S$, $SO_2$ and $SO_3$, soot, carbon monoxide, and up to 3 million tons of carbon dioxide, as well as, which is no less important, the loss of hundreds of millions of cubic meters of hydrocarbons, raw materials for oil and gas chemistry. For example, the qualified processing of 1000 m$^3$ of associated gas produces 820 m$^3$ of dry gas, 200 kg of a wide fraction of light hydrocarbons, and up to 61 kg of stable gasoline [8].

Given the global relevance of these problems, a wide range technologies which make use of sulfurous compounds have been implemented; however, the strengthening of environmental protection requirements dictates the need to create new technologies. These technologies must be highly efficient with a wide range of purified gases, and must minimize environmental damage while maximizing the yield valuable products. Such technologies should also meet the requirements of compactness and ease of process control.

To this end, catalytic methods are the most attractive, as they allow the conversion of highly hazardous hydrogen sulfide into a nontoxic, marketable product, i.e., elementary sulfur. Basic processes for hydrogen sulfide-to-sulfur conversion are the direct oxidation of $H_2S$ into elementary sulfur and the low-temperature reduction of sulfur dioxide.

Due to the relevance of the aforementioned problems, this paper describes attempts to develop and improve the processes of purification and processing of hydrogen sulfide-containing gases. At present, three main categories of methods for cleaning gases from hydrogen sulfide can be distinguished:

- adsorption methods
- absorption methods
- catalytic methods

The general feature of the first two methods is that they are essentially ways to concentrate hydrogen sulfide from a purified gas, and must operate jointly with sulfur production plants using the Claus method. This process is currently the only large-tonnage method which is able to obtain sulfur from highly concentrated hydrogen sulfide-containing gas streams. It is characterized by:

- multistage operation
- insufficient environmental safety, due to the presence of a high-temperature furnace in the technological chain, which is a source of toxic byproducts
- a limited range of applications (thus, it is impossible to treat gases with hydrogen sulfide contents below 20 vol.% or gas streams with flow rates below 1000 Nm$^3$/h).

Therefore, as a supplement or alternative to the Claus process, direct selective catalytic oxidation of hydrogen sulfide to elemental sulfur is currently being explored.

## 2. Direct Selective Oxidation in the Liquid Phase. RedOx Processes

One means by which to purify gases from hydrogen sulfide is oxidation to elemental sulfur using oxygen in solutions of complex compounds of metals with wide variation of the pH of the medium.

The process proceeds at a rapid rate in a wide range of temperatures at pressures of 5–50 atm and provides a high degree of gas purification from hydrogen sulfide. Especially noteworthy are the processes developed by Wheelabrator Clean Air Systems, Inc (Pittsburgh, PA, USA). (ARI–Lo-Cat I®, ARI–Lo-Cat-II®), Shell Oil Company (Houston, TX, USA), and Dow Chemical (SulFerox®), as well as those based on the process of the direct oxidation of hydrogen sulfide in a solution of iron (3+) chelate complexes [9,10].

In SulFerox®, the reagent used was characterized by increased stability and low capital and operating costs. Reagent costs are 80–100$ per 1 t of hydrogen sulfide. Available

data show that in the process of gas purification, up to 50–80% of methyl mercaptan and 30–60% of carbonyl sulfide can be removed from the initial content.

The SulFerox® process uses a new composition of a complexone, which is similar to EDTA (ethylenediamine-tetraacetate). However, the concentration of iron in the absorbent used is significantly higher (up to 3 wt.%) than in Lo-Cat processes (up to 0.5 wt.%, Figure 1) [11]. The first installation put into operation had a capacity of 120,000 m³/day of the gas containing 4.5% hydrogen sulfide and 57% carbon dioxide at a pressure of 20 atm. The largest installation was launched in 1992 in Denver City, Texas. At this unit, 1500 ppmv of hydrogen sulfide in carbon dioxide gas at a pressure of 20 atm was reduced to 20 ppmv. The Sulferox process is currently the object of the greatest amount of research. From 1990 to 1995, Shell designed, built, and constructed more than 20 installations for the cleaning of various technological gases.

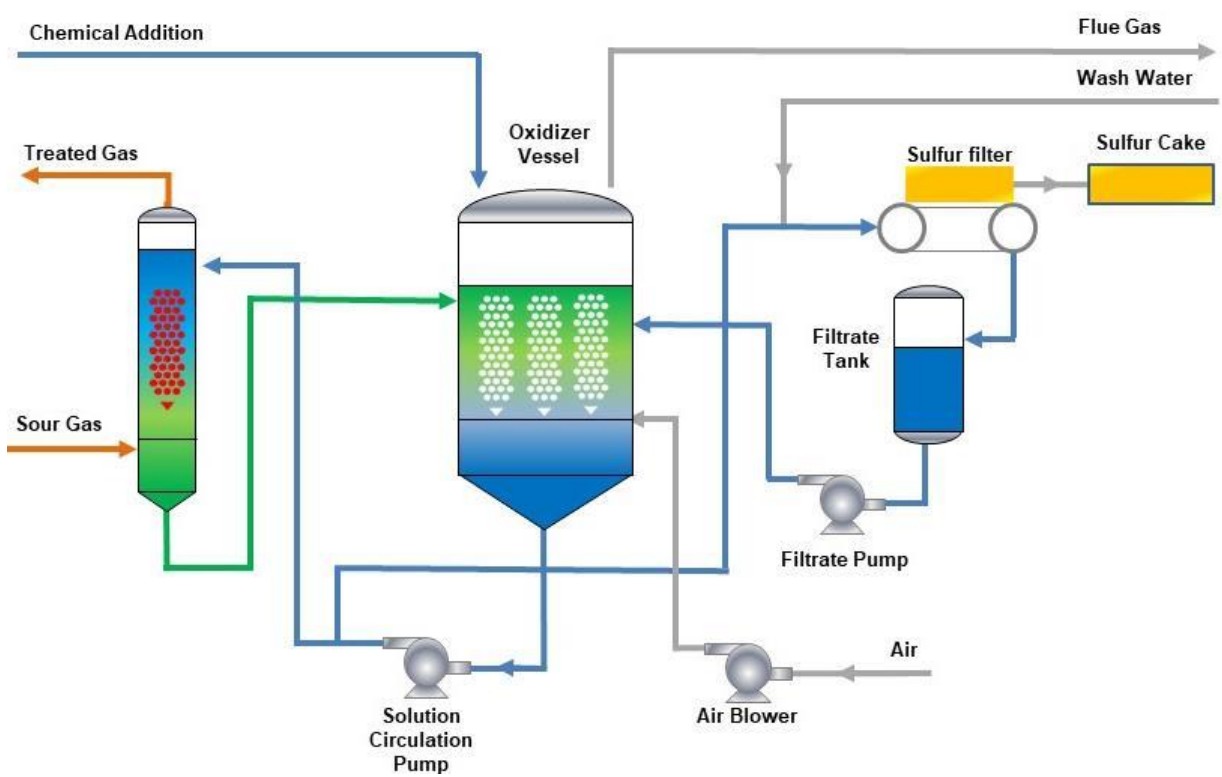

**Figure 1.** Schematic of the Lo-Cat process (adapted from [11]).

In the SulFerox® process, the concentration of iron compounds is significantly higher than in the ARI–Lo-Cat I®, ARI–Lo-Cat-II® processes. This fact explains the broader introduction of the ARI–Lo-Cat processes in gas cleaning operations. In the literature, information was found on the creation of only a few technological complexes for the purification of gases which simultaneously yield elementary sulfur, as opposed to the conventional procedure of amine absorption coupled with the Claus process. The Volga Research Institute of Hydrocarbon Fuels (JSC VNIIUS, Kazan, Russia) developed the Serox-2 process for cleaning gas flows from hydrogen sulfide with solutions of iron complexes to obtain elemental sulfur. The process is an analog of the "LO-CAT" process; its main distinctive feature is the composition of the absorbent with low corrosion activity with respect to carbon steel and high stability under the conditions required for the purification of gases. The process is implemented according to a standard two-step procedure, i.e., sulfur foam filtration and purification of hydrocarbon gas with a residual $H_2S$ content of no more than 20 mg/m³ (National Standard 5542-87) [12].

As more than 40 years of field testing experience shows, the process of cleaning gases from hydrogen sulfide with chelate complexes (iron salts of EDTA) has some disadvantages that limit its application for gases with a hydrogen sulfide content of more than 1–2 g/m$^3$.

Due to the need to use a working solution with a pH no higher than 8.5–9.0, the applicability of this solution with respect to hydrogen sulfide is limited, leading to the need to increase the rate of its circulation through the absorber (i.e., energy consumption for pumping increases).

The formation of a side product in the oxidation of hydrogen sulfide–thiosulfate is inevitable, which necessitates the use of a reagent such as EDTA.

A substantial technological problem is the separation of sulfur from the resulting pulp. Although to date, automatic filter machines (automatic filter presses and drum vacuum filters) have been developed, the complexity of their operation and their high costs make the cleaning process economically costly. Additionally, the low quality of the resulting elemental sulfur makes its commercialization difficult.

### 3. Claus Process

The most widely-used procedure for the large-scale reprocessing of highly-concentrated gases is the Claus method [13], which consists of several steps (Figure 2). The feed for the Claus process is acid gases.

The Claus process is the dominant technology to produce gas (regenerated) sulfur. It is worth noting that the vast majority (about 94%) of the 8.1 million metric tons of sulfur produced in the United States in 2020 was synthesized using the Claus process [14].

The term "acid gases" is used to designate gases obtained after the absorptive treatment of hydrocarbon raw materials. Some typical characteristics of acid gases of various origins are shown in Table 1.

**Table 1.** Typical characteristics of acid gases of various origins.

| Process | H$_2$S Content, Vol.% | Other Gas Components |
|---|---|---|
| Purification of gases from oil processing (MEA treatment) | 90–98 | Carbon dioxide, hydrogen, methane |
| Purification of natural and oil-associated gas (MEA or DEA treatment) | 10–70 | Carbon dioxide, water vapor, hydrocarbons C$_1$-C$_6$ |

As a rule, acid gases formed in the process of hydrotreating oil fractions are characterized by rather low flow rates (≤ 1000 nm$^3$/hour) and high contents of hydrogen sulfide, the concentration of which, depending on the efficiency of the primary cleaning unit, usually exceeds 90%, compared with acid gases produced by a gas processing plant (for example, the power of only one technological line at the Astrakhan GPP is over 15,000 nm$^3$/hour).

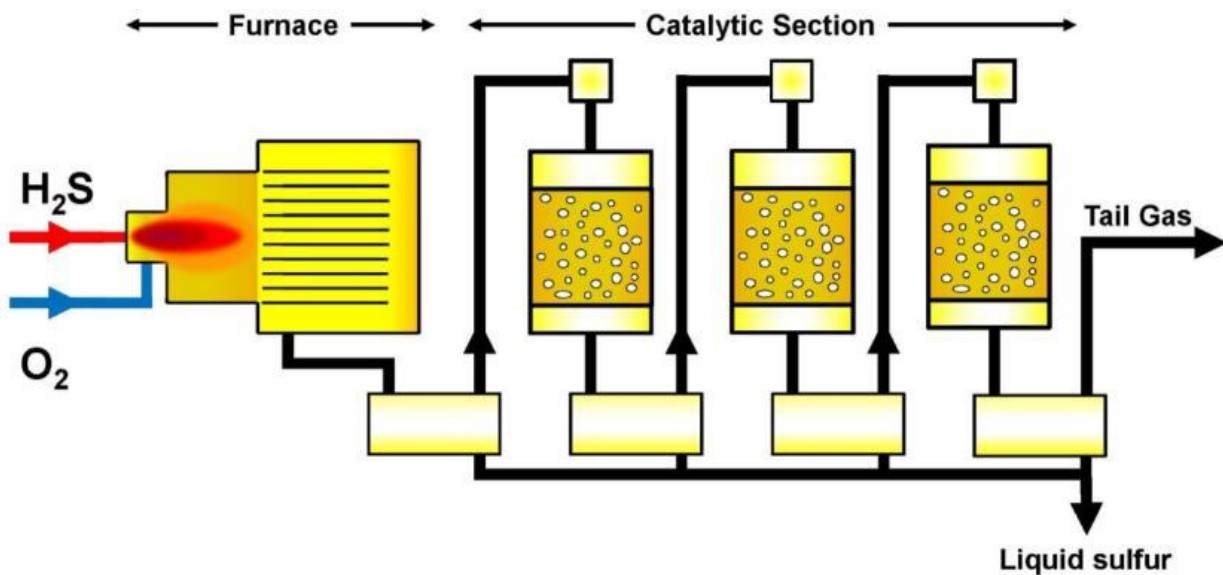

**Figure 2.** Schematic of Claus installation (adapted from [13]).

### 4. The Thermal Stage of the Claus Process. Process Conditions. Chemical Reactions Proceeding in the System

The thermal stage of the Claus process largely determines the efficiency of the process as a whole, because at this stage, the main part (up to 70%) of the target product, i.e., elemental sulfur, is produced.

Upon mixing the acid gas with air (at the same stoichiometric hydrogen sulfide:oxygen ratio used in Reaction 1), a gas stream containing $H_2S$, $O_2$, $N_2$, $CO_2$, $H_2O$, sometimes hydrocarbons, and in some cases $NH_3$, $HCN$, etc. is formed, which is fed to the Claus furnace. Accordingly, during $H_2S$ oxidation in the furnace, in addition to the main reactions [15–17]:

$$H_2S + 0.5O_2 \rightarrow 0.5S_2 + H_2O \tag{1}$$

$$H_2S + O_2 \rightarrow H_2 + SO_2 \tag{2}$$

$$H_2 + 0.5SO_2 \rightarrow H_2O + 0.25S_2 \tag{3}$$

$$H_2S \leftrightarrow H_2 + 0.5S_2 \tag{4}$$

$$CH_4 + 2S_2 \leftrightarrow CS_2 + 2H_2S \tag{5}$$

$$CO + H_2O \leftrightarrow CO_2 + H_2 \tag{6}$$

$$CO + 0.5S_2 \leftrightarrow COS \tag{7}$$

$$CH_4 + 0.5O_2 \rightarrow CO + 2H_2 \tag{8}$$

$$CO + H_2S \leftrightarrow COS + H_2 \tag{9}$$

Since the oxidation of hydrogen sulfide is an exothermic reaction, the temperature in the Claus furnace can reach 1200–1500 °C, while the minimum critical value of the temperature for sustaining a steady flame in the furnace is 1050 °C. The factor determining the temperature of the flame in the standard implementation of the process is the concentration of hydrogen sulfide in the acid gas. Despite significant progress in developing burner devices for the combustion of hydrogen sulfide-containing mixtures, the optimal conditions for the stable operation of the flame furnace are those with a hydrogen sulfide

content in the feed gas of ≥60 vol.%. Technical approaches for maintaining sustainable operation in Claus furnaces are examined in [18] where, along with the results of calculations carried out using the Gibbs energy minimization method, the experimental data are in good compliance with the results of the calculations.

The following methods are considered:

- The reheating of the initial gas streams, acid gas and air: Even at a concentration of hydrogen sulfide in the acid gas of 40 vol.%, it is necessary to heat initial gas streams to 300 °C to reach the lower threshold of the stable operation of the Claus furnace, i.e., 1050 °C. In practice, as the experience of operating Claus installations at the Orenburg GPP shows, considering the essential heat loss, the preheating temperature can be as high as 600 °C.

- The use of oxygen-enriched air as an oxidant: Even at hydrogen sulfide concentrations in the acid gas of 50%, the required oxygen concentration in the supplied air should be at least 50 vol.% in order to reach the lower threshold of the stable operation of the Claus furnace.

- Supply of hydrocarbon fuel gas to the flame furnace: A supply of fuel gas at 25–30% of the acid gas flow rate with a high $H_2S$ concentration will not provide the necessary temperature in the furnace to maintain stable operation. The heat of the combustion of hydrogen sulfide is utilized by heating chemically purified water, with water vapor production in a waste heat boiler. The hot gas passes through the boiler tubes and heats the water therein to boiling point. The gas cooled in the boiler is sent to the condenser, where it is cooled further to about 150 °C.

## 5. Catalytic Stage of the Claus Process. Catalysts Used

Gases from the Claus furnace condenser located after the waste heat boiler containing mainly $H_2S$, $SO_2$, $N_2$, $CO_2$, $H_2O$, COS, $CS_2$, CO, $H_2$, and traces of sulfur are further passed to the main catalytic stage. The process is usually carried out in the adiabatic fixed beds of the granular catalyst, in which, in addition to the Claus reaction, hydrolysis reactions of sulfur-organic compounds also proceed [17,19–22]. Catalytic convertors:

$$2H_2S + SO_2 \leftrightarrow 2H_2O + 0.5S_6 \tag{10}$$

$$COS + H_2O \rightarrow CO_2 + H_2S \tag{11}$$

$$CS_2 + 2H_2O \rightarrow CO_2 + 2H_2S \tag{12}$$

$$2H_2S + SO_2 \leftrightarrow 2H_2O + 0.5S_6 \tag{13}$$

The most commonly used catalyst for the Claus process is aluminum oxide with various modifications. The production of catalysts for the Claus process reaches hundreds of thousands of tons; the big players in this market are BASF [23], producing Claus catalysts Activated Alumina–DD-431, Promoted Alumina DD-831, EURO SUPPORT (previously Kaiser Alumina), and their successors LaRoche and UOP [24], producing catalysts S-2001/ESM-221, S-501/ESM-251, Axens [25] alumina catalysts CR 3-7, CR-400, CR-3S.

A new generation of Claus catalysts based on titanium dioxide [26] is now being actively implemented. The CRS-31 catalyst of French companies Rhone Poulenc and Elf Aquitaine (the current name is Axens Procatalyse, Paris, France) has gained broad recognition. Experience with its industrial use has revealed high stability for a long time in the presence of oxygen, high activity in the Claus reaction, and COS and $CS_2$; see also [25,27].

At the Boreskov Institute of Catalysis SB RAS, the activities of the oxides of various metals in the Claus reaction were comparatively studied [27,28]. Twenty-one oxides were investigated, of which nine were stable. These metal oxides can be arranged according to activity per surface area as follows:

$$V_2O_5 \gg TiO_2 > Mn_2O_3 > La_2O_3 > CaO > MgO > Al_2O_3 > ZrO_2 \gg Cr_2O_3$$

The surface activity of vanadium pentoxide is 16 times higher than that of titanium dioxide and 73 times higher than that of $\gamma$-alumina. However, pure $V_2O_5$ has a low value of specific surface area, and, in connection with the activity per unit of mass, it is inferior to $TiO_2$ and approximately equivalent to $Al_2O_3$. Furthermore, vanadium pentoxide is not very effective in the hydrolysis reaction of sulfur-organic compounds. However, its use in mixed catalytic systems is promising. Based on $V_2O_5$ at the Boreskov Institute of Catalysis SB RAS, the ICT-27-36 catalyst was developed. This catalyst is characterized by high activity in Claus and hydrolysis reactions, high stability at operation in oxygen-containing mixtures, and high mechanical strength [29].

## 6. Claus Process. Enhancement. Oxygen Enrichment

It should be noted that the enriched oxygen in the air supplied in the thermal stage of the Claus process is obtained using COPE® Technology (Kingswinford, UK, The Claus Oxygen Based Enhancement, Figure 3), developed by GOAR, Allison & Associates, LLC [30,31]. This technology is used in installations for sulfur production; its main advantage is the possibility of increasing in the power of the Claus process without incurring significant additional expenses.

Based on experiments and calculations, it was shown that an increase in the concentration of $O_2$ in the air, i.e., to 30 vol.% (a low degree of enrichment), could increase the Claus installation capacity by 25%. It is proposed to transport liquid oxygen in cryogenic tanks without on-site cryogenic or membrane separation installations. This configuration is optimal, giving rise to sulfur production of up to 50 tons/day. The average degree of air enrichment with oxygen ($C_{O2}$ = 40–45 vol.%) will increase the installation capacity by 75%. In this case, the furnace must be equipped with additional nozzles to supply oxygen. The anticipated production of such a plant is 100 tons sulfur/day. With an increase in oxygen concentration up to 100%, the daily production of sulfur can be increased by 150%. However, this will require significant changes in the structure of the flame furnace or the combustion of hydrogen sulfide using the "Sure" Double Combustion Process technology developed by Lurgi [32]. The process is conducted as follows: the burning of hydrogen sulfide with pure oxygen is carried out in a two-section furnace; the reaction products subsequently enter the sulfur condenser and the water condenser, and in recycling mode are then fed to the inlet of the torch.

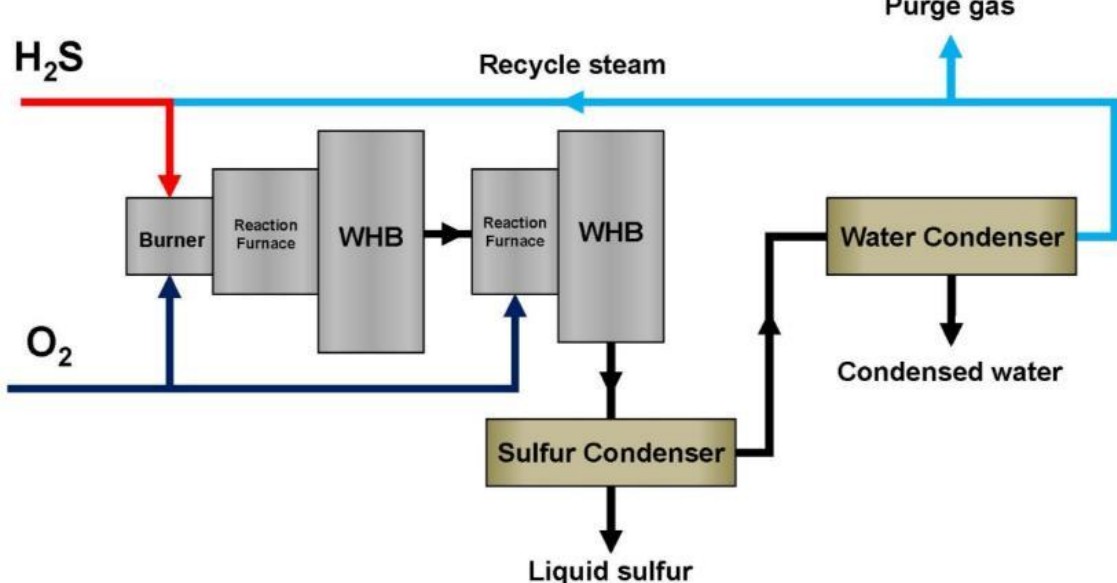

**Figure 3.** COPE® Technology (Kingswinford, UK) flow sheet diagram (adapted from [28]).

It should be emphasized that the cost of reconstruction (capital investment) of existing installations for the transition to the Cope® technology is only 5–25% of the construction cost of a new installation with increased capacity.

## 7. PROClaus Process

In the proposed concept of the modification of the Claus process, the first and second stages are standard: the combustion of hydrogen sulfide of acid gas in a flare furnace with further catalytic conversion of a mixture of hydrogen sulfide and sulfur dioxide in the first catalytic reactor. The main distinguishing feature of the PROClaus process is the use of a specially developed catalyst for sulfur dioxide reduction comprising the oxides of Fe, Co, Ni, Cr, Mo, Mn, Se, Cu, and Zn, in the presence of which, in a temperature range of 200–380 °C, there is almost complete sulfur dioxide conversion into elemental sulfur. Additional reducing agents are the products of side reactions proceeding in the high-temperature Claus furnace, i.e., CO and $H_2$ [33–35].

Furthermore, sulfur is separated from the gas stream containing hydrogen sulfide as the primary reagent, and the gas flows to the direct catalytic oxidation reactor which is filled with a Hi-Activity catalyst [36]. The Hi-Activity catalyst is a modified form of the KS-1 catalyst previously developed in the Azerbaijan Institute of Oil and Chemistry containing iron, zinc, and chromium oxides as the main components [37].

The calculated value of the total extracted sulfur from the gas is 99.2% using the three-reactor scheme and 99.5% for the four-reactor one. There characteristics were confirmed in laboratory studies of the concept of the process. However, the tests at the industrial level ended unsuccessfully: the hypothetical level of sulfur extraction was not observed, as the sulfur dioxide reduction catalyst did not achieve the proposed rate of $SO_2$ conversion into elementary sulfur. According to Alkhazov [38], in the process of laboratory studies, a factor of inhibition of the catalyst activity by sulfur vapor coming with the gas flow after the condenser of the first catalytic stage was not taken into account.

At the same time, according to the company JACOBS [39], the EUROCLAUS process was implemented on an industrial scale using the concept of catalytic reduction of $SO_2$ with the subsequent oxidation of hydrogen sulfide to elementary sulfur. In the EUROCLAUS process, an additional bed of the reduction catalyst is loaded into the Claus catalytic converter.

## 8. SuperClaus Process

The main distinguishing feature of the modification of Claus technology known as the SuperClaus process is the supply of substoichiometric air in the thermal stage (Figure 4). Such a method results in the reaction mixture composition after the second catalytic converter containing predominantly hydrogen sulfide at a concentration of 0.8–3.0 vol.%, with trace amounts of sulfur dioxide. Such a mixture passes to the third sequential reactor filled with a catalyst for the direct catalytic oxidation of hydrogen sulfide [40,41].

According to data provided by Jacobs Comprimo, the technology licensee from the beginning of the first industrial demonstration of the process in 1988, over 190 installations using the SuperClaus® process are currently operating or are under construction, with a total capacity of up to 1200 tons of sulfur per day.

According to the authors, the catalyst provides the sulfur yield in the third converter at a level of 85%, and the total sulfur yield is 99–99.5% [42]. It should be noted that industrial experience shows the inconsistency of the real and expected results. Thus, in the SuperClaus® process, at the stage of the selective oxidation of hydrogen sulfide under industrial conditions, the sulfur yield does not exceed 80–83%, and the achieved total degree of sulfur extraction in SuperClaus® industrial installations is 98–98.6% [43], instead of the declared 99–99.5%. However, a sulfur yield of 85% at the stage of selective oxidation of hydrogen sulfide in treatment of tail gases of the Claus process is currently recognized as the best modern level for technologies using the direct heterogeneous catalytic oxidation of hydrogen sulfide.

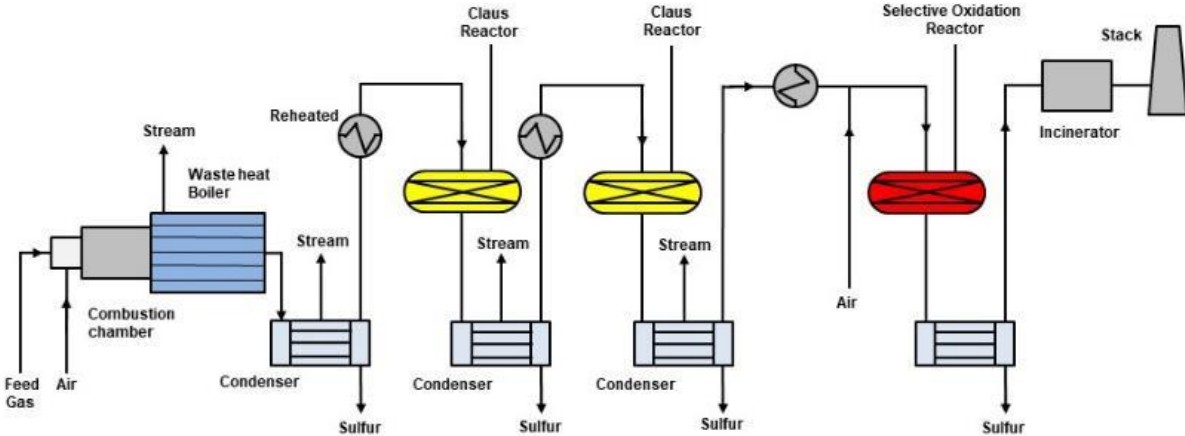

**Figure 4.** SuperClaus® process flow sheet diagram (adapted from [28]).

## 9. Modifications of the Claus Process

Research attempts have been made to optimize the Claus process (notably, the catalytic part). It has been proposed that the interaction of sulfur dioxide and hydrogen sulfide be carried out in the fluidized catalyst bed. Studies were carried out in a cylindrical reactor with an internal diameter of 0.1 m and a height of 0.86 m. Spherical active alumina of the Kaiser S-501 brand with an effective diameter of 195 μm was used as a catalyst. The maximum concentration of the reagents were $H_2S$-1300 ppmv and $SO_2$-650 ppmv, and the test temperatures were 100–150 °C, that is, below the dew point of the sulfur. The fluidization number was varied in the range of 2.2–8.8. It was shown that the observed conversion of sulfur compounds was ~96% in the initial period, although this decreased with an increase in sulfur sediments to 60% in 16 days of continuous operation. As the main advantages of the method, the authors note catalyst loading was reduced by up to 50% compared to the three-reactor scheme of the Claus process. The developed method could be considered as an alternative to the known processes of purification of tail gases based on sulfur condensation (CBA, Sulfreen) followed by the regeneration of catalytic material, and not a fundamentally new process for replacing the catalytic stages of the Claus process [44].

A process for the purification of hydrocarbon gas with hydrogen sulfide contents of 2.3–5 vol.% and carbon dioxide of 3–5 vol.% is proposed. The initial gas also contains from 40 ppmv to 90 ppmv benzene, toluene from 45 ppmv to 220 ppmv, xylene from 20 ppmv to 150 ppmv, carbon sulfoxide (COS) from 25 ppmv to 70 ppmv, heavy hydrocarbons (to $C_{50}$), mercaptans from 15 ppmv up to 50 ppmv. The overall gas processing complex includes the amine treatment installation and the "classic" three-reactor scheme of the catalytic conversion of sulfur dioxide and hydrogen sulfide into elementary sulfur. In the proposed procedure, the gas stream coming from the amine treatment unit is split in a ratio of 75%/25%, and the larger flow enters the first zone, a specially developed furnace, while the smaller stream enters a flushing column after the catalytic converters. In the flushing column, at the interaction of the gas flow components with a caustic soda solution, selective absorption of carbon dioxide occurs, and the stream enriched in hydrogen sulfide flows into the second zone of the thermal stage of the total process line. Such technological approaches are specifically used to extend the lower limit of the range of hydrogen sulfide concentrations in the initial gas stream to 30%. Furthermore, according to the authors' statements, this method is an alternative to the COPE process, while the complex as a whole will ensure the following characteristics of the purified gas: the content of hydrogen sulfide in the purified gas is 4 ppmv, the carbon dioxide content is not higher than 1.7 vol.%; and the content of organic sulfur compounds is not higher than 60 ppmv. It should be noted that the proposed procedure was conceived via computer simulations, and did not undergo any testing on the pilot or experimental levels [45].

There are also proposals to increase the degree of hydrogen sulfide conversion by its removal from the tail gas using reagents based on triazines in order to neutralize residual $H_2S$ [46].

Researchers from Politecnico Di Milano developed a rather interesting concept, i.e., the simultaneous disposal of $H_2S$ and $CO_2$ [47].

With regard to the gasification process of coal, they proposed the joint utilization of acid gas components by reacting carbon dioxide and hydrogen sulfide according to the following Equation:

$$2H_2S + CO_2 \rightarrow H_2 + CO + S_2 + H_2O \tag{14}$$

In this case, carbon dioxide is used as a "soft" oxidizer.

At the same time, Pirola and co-authors [47] demonstrated the results of a comparative analysis, where the superiority of the AG2STM process (Acid gas to SynGas) is shown in comparison to the traditional Claus process.

Thus, several essential problems were solved:

- The generation of additional synthesis gas
- Complete recovery of hydrogen sulfide in the form of elementary sulfur
- The utilization of carbon dioxide.

It should be noted that this work was performed on a computer simulation level, and and that the concept has not undergone laboratory and pilot testing.

## 10. Modern Trends in the Field of Hydrogen Sulfide Treatment with the Formation of Elemental Sulfur. Direct Heterogeneous Catalytic Oxidation of Hydrogen Sulfide to Elemental Sulfur

The process has some significant advantages, the main of which are:

- the single-step characteristic and continuity;
- "soft" conditions (T = 220–280 °C) due to the use of highly active catalysts, which allow for the oxidation of hydrogen sulfide directly in the composition of hydrocarbon.

It should be noted that the apparent advantages of the direct oxidation process are the main reason to consider the technologies using Reaction (15) as an alternative to Claus technology [48,49].

## 11. Chemism of the Process of Direct Catalytic Oxidation of Hydrogen Sulfide

In the process of direct $H_2S$ oxidation, the following reactions can proceed [50]:

$$H_2S + 1/2O_2 \rightarrow H_2O + 1/2S_2 \tag{15}$$

$$H_2S + 1/2O_2 \rightarrow H_2O + 1/6S_6 \tag{16}$$

$$H_2S + 1/2O_2 \rightarrow H_2O + 1/8S_8 \tag{17}$$

$$H_2S + 3/2O_2 \rightarrow H_2O + SO_2 \tag{18}$$

$$H_2S + 1/2SO_2 \rightarrow H_2O + 3/4S_2 \tag{19}$$

$$H_2S + 1/2SO_2 \rightarrow H_2O + 1/4S_6 \tag{20}$$

$$H_2S + 1/2SO_2 \rightarrow H_2O + 3/16S_8 \tag{21}$$

The allotropic form of sulfur $S_2$ is stable in the temperature range of 100–900 °C. The characteristic temperature range for the reaction of direct oxidation of hydrogen sulfide is usually 100–300 °C; in this range, sulfur is present as $S_6$ and $S_8$. The so-called reverse Claus reaction accompanies the oxidation of hydrogen sulfide:

$$3S + 2H_2O \rightarrow 2H_2S + SO_2 \tag{22}$$

At reduced temperatures which are typical for direct oxidation, sulfur chains predominantly consisting of six or eight atoms are formed. With an increase in temperature to 800 °C, hydrogen sulfide oxidation proceeds mainly with the formation of sulfur in the form of $S_2$.

In the temperature range of 25–727 °C, the equilibrium constant of the hydrogen sulfide oxidation reaction with oxygen to elemental sulfur is, on average, 10 orders of magnitude higher than that of the reaction of oxidation with sulfur dioxide. Consequently, the probability of the formation of elemental sulfur in Equations (15)–(18) is higher than by Equations (19)–(21) [50].

The thermodynamic features of the reaction of the direct oxidation of hydrogen sulfide are presented in the form of temperature dependence (Figure 5) [51].

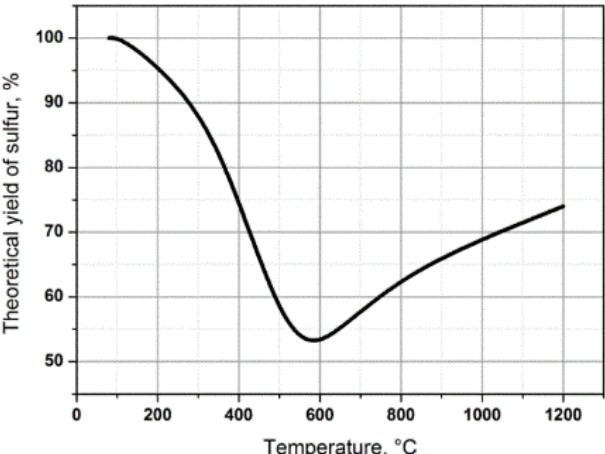

**Figure 5.** The dependence of the equilibrium sulfur yield on the temperature in the reaction of direct oxidation of hydrogen sulfide at atmospheric pressure (adapted from [51]).

The reaction of direct $H_2S$ oxidation can proceed with selectivity achieving 100% target product at low temperatures; with an increase in temperature above 200 °C, the selectivity significantly decreases. The chemical equilibrium is determined by the Claus reaction, i.e., the only reversible reaction of the system. If a catalyst with high activity for the reaction (15) is selected that is practically unaffecting the Claus reaction rate (21), then a super-equilibrium sulfur yield (100%) can be attained [52]. Therefore, the use of $TiO_2$- and $Al_2O_3$-based catalysts in this process is ineffective. An increase in pressure in the system favorably affects the yield of elemental sulfur and increases the selectivity, even at elevated temperatures.

## 12. Main Types of Catalysts Used in the Process of Direct Heterogeneous Oxidation of Hydrogen Sulfide. Industrial Processes. Brief Description of the Most Common Catalysts for the Hydrogen Sulfide Oxidation Reaction with Oxygen to Elementary Sulfur

Specific requirements associated with the particular features of the reaction are imposed on catalysts for the process of the direct oxidation of hydrogen sulfide into elemental sulfur. In terms of selecting a catalyst, the thermodynamics of the process should be taken into account, as well as the possibility of the homogeneous evolution of the process through the radical-chain mechanism at elevated temperatures and the condensation of sulfur in catalyst pores at low temperatures.

Catalysts for the direct oxidation of hydrogen sulfide to sulfur are used in the temperature range of 200–350 °C. The sulfur dew point determines the lower limit of the temperatures. The upper limit is due to the possibility of the reactions of sulfur and hydrogen

sulfide oxidation to sulfur dioxide, which leads to a significant drop in the reaction selectivity.

Despite considerable efforts devoted to the direct catalytic oxidation of hydrogen sulfide described in the literature, the scope of catalytic systems for this process is somewhat limited. These, first of all, are activated carbons and artificial zeolites [53,54], as well as natural bauxites, traditionally used as catalysts for this process [55].

However, the most promising systems are individual metal oxides or mixtures of transition metal oxides due to their apparent advantages, i.e., high mechanical strength, thermal stability, and relative cheapness. It should be noted that oxides are used both in a bulk state and in the supported form. This is confirmed by the fact that all commercial processes for sulfur production from $H_2S$ through its direct catalytic oxidation, such as Catasulf® of BASF (Ludwigshafen, Germany), BSR/SELECTOX® of Unocal Company (California, CA, USA), Modop® of Mobil Oil (Panama City, Panama), etc., are based on the application of heterogeneous multicomponent oxide catalysts.

### 13. Activated Carbon. Catalysts Based on Activated Carbon

As demonstrated above, active carbon (AC) simultaneously acts as the adsorbent of hydrogen sulfide and the catalyst for the oxidation of the latter to sulfur, as $H_2S$ is transformed into sulfur which accumulates in AC pores upon purification.

Microporous ACs have been well investigated as adsorbents/catalysts for the periodical partial oxidation of hydrogen sulfide at temperatures below 150 °C [56–66]. They demonstrate high activity and selectivity under these conditions. As shown, a relatively large volume of large pores is required for the oxidation process to occur, whereas smaller pores serve for adsorptive desulfurization processes. Elemental sulfur is mainly accumulated in pores <12 Å, and initially in small and later in larger ones. It has been shown that there is maximum adsorption in cases where the pore size is maximally close to the size of the adsorbent molecules [67,68]. Hence, efficient carbon materials should have the optimal pore structure with a good volume of both micro- and meso- pores, and a relatively narrow pore size distribution to ensure high selectivity for sulfur. Nevertheless, the complete picture of the effect of AC pore structure on $H_2S$ selective oxidation is not quite clear yet.

Primavera and co-authors [69] investigated the effect of water vapor on adsorbent efficiency. It was found that a relative moisture content of about 20% facilitates an enhanced reaction rate. The reaction rate drops dramatically when the moisture content is decreased, and less significantly when increased. It is assumed that $HS^-$ ions are generated in water, being readily oxidized to sulfur with oxygen.

Surface chemistry has a significant effect on catalyst efficiency; therefore, AC-based sorbents/catalysts undergo modification with various reagents, such as metal salts [70] and alkaline [63,64,71] or oxidative (permanganate) additives [62], by the introduction of heteroatoms, such as nitrogen, oxygen, and phosphorus [72], and also by thermal treatment and controlled surface oxidation [73–75].

When AC is treated with nitric acid, oxygenated groups (C=O, C–O, and C–O-) are generated. Modified ACs contain charged oxygen particles, have higher catalytic activities, and may oxidize to 1.9 g of $H_2S$ per g of catalyst, which is much higher than literature data for carbon catalysts [58,62,76,77].

The dynamic adsorption capacity of AC is reduced, as high temperatures decrease adsorption efficiency and selectivity for sulfur because COS and $SO_2$ are formed. In order to improve the capacity for sulfur and catalytic activity at high temperatures, AC modified with metal oxides is used [62]. At 180 °C and in the absence of water vapor, catalytic activity is varied in series, i.e., Mn/AC > Cu/Ac > Fe/AC > Ce/AC > Co/AC, being reduced to between 142 mg and 6 mg of $H_2S$/g for Mn/AC and V/AC, respectively. The major reaction product is elemental sulfur, which forms on active sites (carrier and coal micropores). When these pores are blocked with sulfur, the catalyst is deactivated.

When CO and $CO_2$ are present in the gas, a side product, COS, appears [78,79]. The impregnation of AC with sodium hydroxide facilitates hydrogen sulfide conversion [71,80–82], as NaOH improves $H_2S$ dissociation to form hydrosulfide ion ($HA^-$) followed by its oxidation to S, $SO_2$, and $H_2SO_4$. Hydroxyl groups ($OH^-$) on the carbon surface enable binding $SO_2$ with COS due to an ion-dipole interaction between $OH^-$ and COS.

Reaction conditions for the selective catalytic oxidation of hydrogen sulfide (temperatures, ratios of $O_2/H_2S$, and volumetric flow rate) have a significant effect on the activity and selectivity of AC-based catalysts [61]. Herewith, the microporosity and relatively small pore volume limit adsorptive capacity, with values of 0.2–0.6 g of $H_2S$/g for AC treated with alkalis and 1.7–1.9 g of $H_2S$/g for AC with oxygenated groups on the surface; sulfur saturation of the catalyst requires its frequent replacement. Other drawbacks of such adsorbents/catalysts preventing their wide applications are connected with the trend to spontaneously ignite upon hydrogen sulfide adsorption on alkaline AC and limited regeneration possibilities.

Sun and co-authors [83] describe the synthesis and properties of nitrogen-doped mesoporous carbon. This material shows a high concentration of catalytically active sites and a large pore volume. When nitrogen content is 8%, adsorptive capacity values of 2.77 g of $H_2S$/g at 30 °C and relative moisture content of 80% were achieved. The presence of pyridine nitrogen explains the elevated capacity. Nitrogen atoms located at the facets of graphite cavities have a high electron acceptor capacity, which facilitates the adsorption of oxygen atoms and therefore facilitates the oxidative reaction. Furthermore, the presence of pyridine active sites on the surface increases the basicity of the aqueous layer therein and simplifies $H_2S$ dissociation to form $HS^-$ ions. The nitrogen content plays a key role, affecting the basicity and thus the concentration of $HS^-$.

## 14. Catalysts Based on Carbon Nanotubes

Nanocarbon materials, i.e., carbon nanotubes (**CNT**) and carbon nanofibers (**CNF**), have recently attracted considerable research attention [84–86]. In particular, due to their lack of microporosity, diverse structures (the outer or inner diameter and the number of graphene layers), and rich surface chemistry (heteroatoms and structural defects), they are more promising than microporous AC, in which quite a few micropores substantially increase the role of diffusion. In particular, the tubular morphology of CNT ensures a special reactivity with liquid and gas reagents when passing through small tubes. For example, the so-called confinement effect [84] should be mentioned. Moreover, the chemical inertness of CNT species avoids problems of sulfation.

Metal oxides, alkaline agents, and heteroatoms are often used to modify CNT. According to the data of Nhut and co-authors [87], $Ni_2S$-modified CNT has a high capacity for sulfur (1.8 g of $H_2S$ per catalyst) in a trickle-bed reactor. Active sites of $Ni_2S$ are located inside the tube due to the confinement effect, and condensed water acts as the conveyor track, transferring elemental sulfur from the inner graphene layers to the outer ones in multilayer CNT, from where it is desorbed from the active phase. This mechanism ensures a high rate for hydrogen sulfide removal without any deactivation for 70 h. A substantial free catalyst volume makes it possible to save the resulting sulfur. However, because of the hydrophobic properties of $Na_2S$/CNT, condensed water is required to maintain high activity, which complicates reactor design and production.

Multiwalled CNT modified with $Na_2CO_3$ also make it possible to achieve capacity values of 1.86 g of $H_2S$/g catalyst at a low temperature (30 °C), which is approximately four times higher than commercial AC (0.48 g of $H_2S$/g catalyst) [88]. As in the case of $NiS_2$/CNT, a high capacity for sulfur is ensured by the presence of a large free volume formed by voids between CNT aggregates. In addition, introducing $Na_2CO_3$ increases the hydrophilicity and alkaline properties of CNT as an adsorbent. Alkaline properties promote the sorption and dissociation of $H_2S$ into $HS^-$ ions in the aqueous layer. The gradual deactivation of the catalyst is linked to a decrease in pH upon sodium sulfate formation and the blocking of catalyst pores with sulfur.

Hydrogen sulfide oxidation over multiwalled CNT decorated with tungsten sulfide was investigated in [89,90]. The metal content in the catalysts was 4.7–4.9%. The catalyst activity was examined compared to $WS_2/AC$ and $WS_2$ catalysts using single-walled CNT under the following conditions: 5000 ppmv of $H_2S$, 20% of water vapor, a volumetric flow rate of 5000 $h^{-1}$, $O_2/H_2S = 2$, and a temperature of 60 °C. As shown, the catalyst over multiwalled tubes displayed the highest activity. The catalyst activity has been shown to increase with increased metal content but to cease when the latter is over 15%. When the volumetric flow rate is increased, the conversion degree naturally decreases. Upon an increase in temperature to between 70 °C and 180 °C, there is a high degree of conversion of hydrogen sulfide (at 180 °C), i.e., close to 10%, which is stable for 10 h, in contrast to the process performed at lower temperatures. This is related to the fact that sulfur is removed from catalyst pores more quickly at a high temperature, i.e., close to the melting point.

Macroscopical nitrogen-doped CNT (N-CNT) were developed by Ba al [91] for hydrogen sulfide oxidation at high temperatures (>180 °C) with heavy mass flow rates, WHSV = 0.2 − 1.2 $h^{-1}$. As demonstrated, $H_2S$ conversion increases with nitrogen content, which is associated with a simultaneous increase in the concentration of active oxygen sites. Correspondingly, when the temperature was 250 °C, the degree of $H_2S$ conversion and selectivity were 91% and 75%, respectively. When the catalyst is deposited onto a spongy carrier, SC, process indicators are substantially improved: conversion degree and selectivity reach 90% after 120 h of operation at 190 °C and high WHSV values.

A recent paper by Chizari and co-authors [92] investigates the activity of N-CNT catalysts formed as spherical granules with a diameter of around 5 nm. The test conditions were as follows: temperature of 210–230 °C, $H_2S$ concentration of 1%, $O_2$ content of 2.5%, water level of 30%, and gas hourly space velocity (GHSV) of 2400 $h^{-1}$. As shown, the N-CNT catalyst was more efficient in terms of hydrogen sulfide removal compared to the $Fe_2O_3/SiC$ catalytic agent: the conversion degree reached 100% and selectivity was around 80% at 210 °C compared to the deposited oxide catalyst, for which $H_2S$ conversion degree under these conditions was only 30%.

## 15. Carbon Nanofiber-Based Catalysts

As in the case of nanotubes, the main advantages of carbon nanofiber-based catalysts are related to the high thermal conductivity of the latter, chemical inertness, and the lack of ink-bottle pores where elemental sulfur may settle [93]. Furthermore, the presence of pores as microcavities between nanofibers increases the sulfur capacity of the material.

Using CNT for $H_2S$ selective oxidation at high temperatures (>180 °C) has been investigated more widely than CNT-based catalysts. The latter are more promising from the standpoint of using a high excesses of oxygen to stoichiometry [94]. Herewith, the catalytic characteristics might be quite different depending on the nature of the initial catalyst over which the synthesis of nanofibers was carried out.

When water is absent, nanofibers produced over a Fe-Ni catalyst [95] with a structure of multilayered CNT have the highest selectivity for sulfur. The selectivity for sulfur is maintained at a level of 90%, whereas $H_2S$ conversion degree decreases to 65% after 25 h of the reaction. The most highly active CNT samples were obtained using a Ni-Cu catalyst. After 25 h of reaction, hydrogen sulfide conversion degree and selectivity for sulfur were 95% and 70%, respectively. Compared to those species, nanofibers grown on Ni-catalyst displayed low activity because of sulfur deposits. In order to improve catalytic characteristics, these fibers were modified by treatment with $HNO_3$ or $NH_3$ [91]. As determined, acid treatment improved catalyst stability and selectivity for sulfur due to the the partial removal of nickel from CNF. In contrast, ammonia treatment reduced selectivity. As noted, the presence of 40% of water vapor improved the characteristics of the procedure, achieving a conversion degree of 70% and a selectivity of 89%.

Shinkarev and co-authors [96] investigated the process kinetics of selective hydrogen sulfide oxidation over CNT. The proposed kinetic model matched well with experimental results across a broad temperature range (155–250 °C) with hydrogen sulfide, oxygen, and

water vapor concentrations of 0.5–2 vol.% and 0.25–10 vol.%, and 0–35 vol.%, respectively. The findings may be used when modeling processes and reactor designs for $H_2S$ selective oxidation using nanofiber-based catalysts.

Chen and co-authors [97] systematically investigated $H_2S$ selective oxidation over acrylonitrile-derived CNF impregnated with $Na_2CO_3$. The capacity for sulfur over these catalysts was shown to be 0.10–0.81 g of $H_2S$/g. First of all, the pore structure affected the sulfur capacity. Additionally, unlike other nitrogen-doped carbon materials, the concentration of nitrogenated functional groups almost did not affect the characteristics of the $H_2S$ oxidation process. As demonstrated by analysis data, the prevalent product, i.e., elemental sulfur, was deposited in larger pores, whereas $H_2SO_4$ was generated in smaller ones.

The effect of temperature and water on $H_2S$ selective oxidation over CNF-based macroscopic catalysts was analyzed by Coelho and co-authors [98]. Carbon nanofibers were grown over a graphite fiber substrate. The active phase was $NiS_2$. The catalyst demonstrated very high selectivity and stability at 60 °C owing to its stability to sulfur deposits removed therefrom through the presence of water and the hydrophobic properties of the catalyst. The efficiency of $H_2S$ removal using catalysts based on new nanocarbon materials, i.e., CNT and CNF, was shown to be much higher, and material doping with nitrogen improved the purification process characteristics to a greater extent.

This research demonstrates that carbon materials are highly efficient during direct $H_2S$ oxidation and the sorption of sulfur compounds.

Liu and co-authors [99] described the synthesis and study of a catalyst for the catalytic oxidation of $H_2S$ to S at room temperature. The catalyst was activated carbon with supported iron and cerium oxides. The introduction of ceria was a positive factor, increasing catalytic activity due to the improved oxidation of $Fe^{2+}$ to $Fe^{3+}$ by redox-pair $Ce^{4+}/Ce^{3+}$. Also, the sorption capacity increased significantly. The adsorption-catalytic parameters of the system were investigated at a relative humidity of 80%, an oxygen content of 10 vol.%, a temperature of 30 °C and a space velocity of 7440 $h^{-1}$. The time of continuous stable operation with the sulfide conversion close to 100% was 71 h, and the value of the adsorption capacity was 820 mg S/g catalyst, which significantly exceeded this indicator for KNa/AC systems. It was found that the obtained sulfur is mainly precipitated inside the pore volume of the AC, but that some also formed on the AC surface.

Note that, depending on the nature of the process occurring on activated carbon during gas purification, the requirements for its porous structure may be different. For an adsorption process, carbons with narrow pores are required, the surface of which should have minimal catalytic activity. For catalytic oxidation of hydrogen sulfide, a wide-pore carbon is needed with a large total pore volume and, naturally, high catalytic activity. Large pores are needed to accumulate the resulting sulfur, in which up to 120% sulfur relative to the mass of the carbon can be adsorbed [100].

In connection with the development of technologies for the production of nanoscale carbon fibers, recently, the use of these materials and catalysts based on them in the reaction of partial oxidation of hydrogen sulfide to sulfur has attracted much interest [87,94,101]. It has been shown that carbon nanofibers make it possible to increase both the catalyst activity and sulfur resistance to deposition on the catalyst surface at low temperatures compared to conventional catalysts.

The issue of the use or regeneration of spent AC sorbents/catalysts deserves special consideration. Standard (industrial) processes for solving this problem are:

1. burning out sulfur at elevated temperatures
2. treatment of catalyst/sorbent with steam, with resulting hydrogen sulfide formation
3. washing catalyst with an organic solvent, effectively dissolving sulfur.

Obviously, the second and third options are the most acceptable for carbon materials, because, when exposed to oxygen at high temperatures, destruction (combustion) of the carbon matrix will inevitably occur.

### 16. Zeolite Catalysts for Direct Oxidation of Hydrogen Sulfide

Along with activated carbon, zeolites also can be used as adsorbents with catalytic properties for the oxidation reaction of hydrogen sulfide with molecular oxygen [102,103]. For low concentrations of hydrogen sulfide, the activity of zeolites NaX, NaY, NaA decreases with time; however, after a certain time (stabilization time), the fall in activity is terminated. The stabilization time decreases with a decrease in the hydrogen sulfide concentration in the gas and does not depend on the temperature of the process. At temperatures below 300 °C, the degree of transformation of $H_2S$ does not depend on its initial concentration.

The oxidation of hydrogen sulfide on various zeolites was detailed studied in detail by Z. Dudzik and co-authors. In [103], the reaction of the direct oxidation of hydrogen sulfide on the sodium form of faujasite was examined. As shown, when the oxygen pulse is supplied to the activated zeolite NaX, on which hydrogen sulfide was preadsorbed, the sample became intensely paramagnetic, and the electronic paramagnetic resonance method allowed the registration of a sulfur biradical -*S-(S)-S*. The measurement of catalytic activity showed that sodium faujasite is an effective catalytic system for direct catalytic oxidation of hydrogen sulfide to elementary sulfur, partially paramagnetic sulfur. The degree of hydrogen sulfide removal from the initial gas flow gradually drops and reaches a constant value during the reaction. In this case, the level of stationary activity is directly proportional to the temperature of the process to temperatures of about 150 °C; the further temperature rise leads to the intensive formation of an unwanted by-product-sulfur dioxide.

Lee and co-authors [104], studied oxides of transition metals supported on NaX zeolite as catalysts for the oxidation of hydrogen sulfide in gaseous products of coal gasification. Coal gasification gases can contain $H_2$, $CO$, $H_2S$, $CO_2$, $O_2$, and $H_2O$. The authors identified the influence of the nature of transition metal oxides on the catalyst activity and selectivity to sulfur formation.

It was shown that the catalyst based on vanadium oxide showed the maximum activity ($X_{H2S} = 70\%$) and selectivity ($S = 80\%$). To characterize the composition of coal gasification gases [104–106] introduced the term "Reducing Power" [Equation (I)]:

$$Reducing\ Power = \frac{[H_2S] + [CO] + [H_2]}{[O_2] + [H_2O] + [CO_2]} \tag{23}$$

The strong dependence of the activity and selectivity of catalysts on the "Reducing Power" of coal gasification gases was established. In their work, the authors concluded that the vanadium oxide catalyst could be effectively used to remove $H_2S$ from gases of coal gasification.

### 17. Catalysts Based on Sic

Recently, catalytic systems based on new materials are being intensively developed. One of these materials is SiC-silicon carbide. Catalysts supported on silicon carbide are proposed to be used in highly exothermal reactions such as partial oxidation of hydrogen sulfide to sulfur. Recently, several works were published offering catalysts-metal oxides on silicon carbide [105–107].

The use of SIC as support of hydrogen sulfide oxidation catalysts has several advantages:

1.  The chemical inertness of the material allows the use of catalysts in aggressive media, providing high stability of catalysts;
2.  High SiC thermal conductivity (150 W/m • K) compared to alumina (15 W/m • K) ensures a uniform temperature distribution in the catalyst bed and prevents local overheating of the catalyst;
3.  SiC-based catalysts can be used to remove $H_2S$ from highly concentrated gases (>2 vol.%);

4. The meso- and macroporous SiC structure allows the use of catalysts for the oxidation of hydrogen sulfide at temperatures below the dew point or in the presence of excess water.

Nguyen and co-workers [105] investigated $Fe_2O_3$-based catalysts supported on $\gamma$-$Al_2O_3$ and SiC in the oxidation reaction of 1 vol.% $H_2S$ in the presence of 30 vol.% $H_2O$. It was shown that the SiC catalyst had much higher activity in the hydrogen sulfide oxidation reaction compared with the alumina-based catalyst.

The $Fe_2O_3$/SiC catalyst showed high activity in the hydrogen sulfide oxidation reaction and selectivity to the formation of elemental sulfur in excess of oxygen and in the presence of water vapor. To determine the nature of the active component, $Fe_2O_3$/SiC, $FeS_2$/SiC, and $FeSO_4$/SiC catalysts were synthesized [107]. It was shown that at $H_2S$ conversion close to 100%, the selectivity to sulfur on these catalysts decreases in the following sequence:

$$FeSO_4/SiC > Fe_2O_3/SiC > FeS_2/SiC$$

The catalyst containing sulfate groups on the surface showed selectivity to sulfur of about 100% at 240 °C, whereas the formation of $SO_2$ was observed on the other catalysts in noticeable quantities. For example, on the $FeS_2$/SiC catalyst, the selectivity to sulfur was about 60%.

Keller and co-authors [106] proposed to use for the oxidation of hydrogen sulfide in the presence of water vapor $NiS_2$/SiC catalyst supported on mesoporous SiC. To avoid the catalyst deactivation in the reaction conditions, it was proposed to use a binary catalyst containing the hydrophobic SiC support and the hydrophilic layer of $SiO_2$ located in the support pores of the carrier. The transformation of the initial $NiS_2$ to nickel oxysulfide which has high activity in the hydrogen sulfide oxidation reaction explains the high activity of the proposed catalyst.

The mechanism of the catalyst deactivation in the absence of water vapor was proposed and an explanation of the high stability of the catalyst in the presence of water was found (Figure 6). According to the authors, the catalyst has a hydrophilic layer in SiC pores. Under the reaction conditions, in the presence of water vapor, the water film is formed on the hydrophilic layer, which delivers/transfers the resulting elemental sulfur to hydrophobic parts of the SiC support, where its deposition and the subsequent transition to the gas phase occurs. Thus, the active component remains available for reagents, and the catalyst is not deactivated (Figure 6a).

In the reaction medium without vapor water, such a film is not formed. Therefore, sulfur deposition occurs mainly on the active component of the catalyst, which leads to the capsulation of the active component and deactivation of the catalyst (Figure 6b).

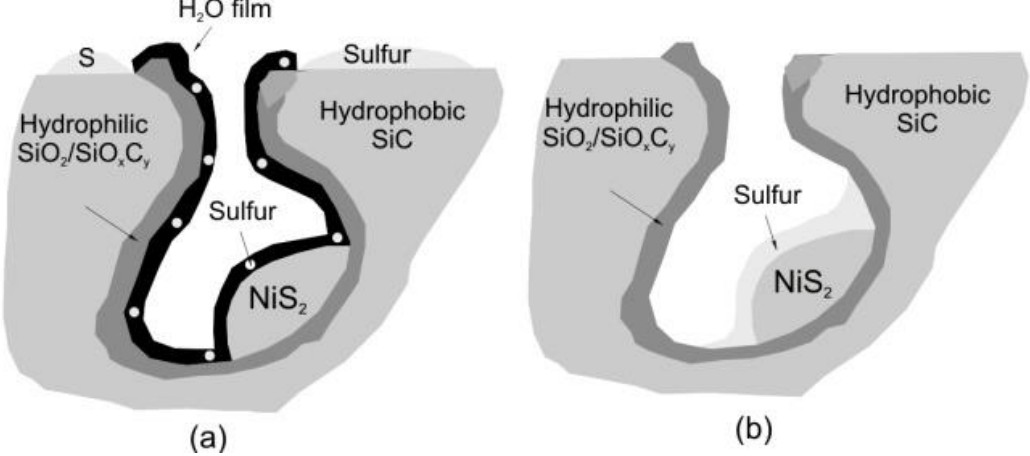

**Figure 6.** The tentative mechanism of sulfur deposition on the surface of a catalyst based on SiC: (**a**) in the presence of water vapor; (**b**) without water vapor (adapted from [106]).

Thus, SiC catalysts are promising for use in the reaction of partial oxidation of $H_2S$ to sulfur. Several SiC-based catalysts were proposed. These are mainly iron oxide systems and a nickel sulfide-based catalyst. However, these catalytic systems are not optimal for the hydrogen sulfide oxidation reaction, and the development of the composition of an active component simultaneously active and selective in the reaction required additional research. The most optimal catalytic systems for the oxidation reaction of hydrogen sulfide to sulfur are transition metal oxides or a combination of oxides.

## 18. Transition Metal Oxides

Catalysts based on metal oxides are most widely used and studied in the continuous process of $H_2S$ selective oxidation. Their main feature is that they provide a stable operation with different $H_2S/O_2/H_2O$ ratios. Also, undoubted advantages of oxide catalysts are high mechanical and thermal stability, availability, wider ranges of hydrogen sulfide concentrations, and space velocities; therefore, their productivity is much higher than that, for example, of carbon-based catalysts [108].

This group of catalysts is promising for the direct oxidation of hydrogen sulfide due to the previously indicated reasons: high mechanical and thermal stability and availability.

Catalysts for gas-phase oxidation of hydrogen sulfide to elementary sulfur are known on bulk alumina or alumina with additives of titania (5.0–15.0 wt.%). The catalysts have high activity, and selectivity in a temperature range of 160–230 °C: the total conversion of hydrogen sulfide into sulfur and sulfur dioxide is 80–100% depending on the temperature range studied [109].

In the practice of gas-phase oxidation of hydrogen sulfide with air oxygen to elementary sulfur, the use of titania in the form of a mixture of its rutile (5–50 wt.) and anatase (50–95 wt.%) modifications as a catalyst is described. In the presence of the catalyst of the specified composition, it is possible at a space velocity of 3000 $h^{-1}$, a temperature of 230–280 °C, an initial $H_2S$ concentration of 3 vol.% with the stoichiometric $H_2S/O_2$ ratio to provide ~98–100% conversion of hydrogen sulfide [110]. However, the catalyst of the specified composition has extremely low mechanical strength. The introduction into the catalyst of strengthening additives of magnesium oxide in an amount 0.3–1.0 wt.% and alumina slightly increases the mechanical strength (the catalyst attrition rate decreases by two times). More significant strengthening (4 to 6 times) is achieved by deposition of the active component on the faience aluminosilicate support.

The complete kinetic data relating to the oxidation of hydrogen sulfide to elementary sulfur on metal oxides can be found in the works of V.I. Marshnyova and Davydov A. A. [111], who studied more than twenty individual metal oxides under the following standard conditions for all samples:

$$T = 250 \text{ °C}$$

The concentration of reagents:

$$C_{H2S} = 0.5 \text{ vol. \%}$$

$$C_{O2} = 0.25 \text{ vol. \%}$$

It was shown that for the kinetic region, an activity row of individual oxides is as follows:

$$V_2O_3 > Mn_2O_3 > CoO > TiO_2 > Fe_2O_3 > Bi_2O_3 > Sb_6O_{13} > CuO > Al_2O_3 = MgO = Cr_2O_3$$

Representing the conversion of hydrogen sulfide in the form of three reactions: the Claus reaction (I), the total oxidation reaction (II), and the reaction of partial oxidation (III) [Equations (22)–(24)], Alkhazov and coauthors [112] found the following patterns for individual oxides.

$$2H_2S + SO_2 \leftrightarrow 2H_2O + 3/nS_n \tag{24}$$

$$2H_2S + 3O_2 \rightarrow 2H_2O + 2SO_2 \tag{25}$$

$$2H_2S + 3O_2 \rightarrow 2H_2O + 2/nS_n \tag{26}$$

It was shown that for the kinetic region, an activity row of individual oxides is as follows:

$V_2O_5 > Mn_2O_3 > CoO > TiO_2 > Fe_2O_3 > Bi_2O_3 > Sb_6O_{13} > CuO > Al_2O_3 = MgO = Cr_2O_3$

Whereas for the total conversion of hydrogen sulfide to sulfur and sulfur dioxide (II + III) these oxides can be arranged in the following row [111]:

$V_2O_5 > Bi_2O_3 > Fe_2O_3 = CoO > Mn_2O_3 > Sb_6O_{13} = TiO_2 > CuO > Cr_2O_3 > Al_2O_3 > MgO$

The maximum stationary activity in all reactions (I-III) is observed for vanadia $V_2O_5$. Analyzing the data on selectivity, Davydov and co-authors [111] concluded that the most selective catalysts for the process (III) are $V_2O_5$, MgO, and $Mn_2O_3$, while the oxides of Bi, Fe, and Cu are catalysts for deep oxidation of hydrogen sulfide to $SO_2$ (II).

The above series of activities are significantly different from similar ones given by T.G. Alkhazov and N.S. Amirgulyan [37] who studied the catalytic properties of metal oxides of the IV period to select the optimal catalyst for the partial oxidation of hydrogen sulfide. According to their data, the catalytic activity of individual oxides in the reaction of direct selective oxidation of hydrogen sulfide to elementary sulfur at temperatures of 280–300 °C decreases in the following sequence:

$Co_3O_4 > V_2O_5 > Fe_2O_3 > Mn_2O_3 > CuO > TiO_2 > ZnO > NiO > Cr_2O_3$

They also give the activity row of these oxides in the reaction of deep $H_2S$ oxidation to sulfur dioxide:

$Co_3O_4 > V_2O_5 > NiO > ZnO > CuO > Fe_2O_3 > TiO_2 > Cr_2O_3 > Mn_2O_3$

Unfortunately, it is impossible to determine the causes of discrepancies, since in [39], absolute values of the specific rates and the conditions for conducting experiments are not given: the ratio of reagents, the size of the catalyst pellet, etc.

Batygina and co-authors [113], studied the catalytic activity of transition metal oxides deposited on $\gamma$-$Al_2O_3$ under the conditions listed below:

- The content of hydrogen sulfide in the feed, vol.%    20;
- Gas hourly space velocity, $h^{-1}$     7200;
- Hydrogen sulfide/oxygen polar ratio     2/1;
- Temperature range of testing, °C    200–300;
- The geometric shape of catalysts    spherical granules;
- Active component    individual oxides of cobalt;
- manganese, chromium;
- Vanadium;
- The active component content, wt.%     0.1–0.6.

It was shown that with the stoichiometric ratio of reagents, the activity of metal oxides decreased in the following sequence:

- $Co > V > Fe = Cr > Mn > \gamma$-$Al_2O_3$ (at T > 250 °C);
- $V > Fe = Cr > Co > Mn > \gamma$-$Al_2O_3$ (at T < 250 °C).

$CeO_2$-based catalysts are potentially suitable for $H_2S$-selective oxidation, but their practical application is limited due to the problem of sulfate formation. Shape-specific $CeO_2$ nanocrystals (rods, cubes, spheres and nanoparticles) with well-defined crystal facets and hierarchically porous structure were successfully synthesized and used as model catalysts to study the structure-dependent behavior and reaction mechanism for $H_2S$

selective oxidation over ceria-based catalysts. It is deduced that the defect sites and base properties of $CeO_2$ are intrinsically determined by the surface crystal facets. Among the nanocrystals, $CeO_2$ nanorods with well-defined [110] and [100] crystal facets exhibits superb catalytic activity and sulfur selectivity. The high reactivity for $H_2S$ selective oxidation is attributed to the high concentration of surface oxygen vacancies which are beneficial for the conversion of lattice oxygen to active oxygen species. Besides, the presence of hierarchically porous structure of $CeO_2$ nanorods hinders the formation of $SO_2$ and sulfate, ensuring good sulfur selectivity and catalyst stability. Through a combined approach of density-functional theory (DFT) calculations and in situ DRIFTS investigation, the plausible reaction mechanism and nature of active sites for $H_2S$ selective oxidation over $CeO_2$ catalysts have been revealed. Thus, morphology engineering can be one of the effective methods in boosting the $H_2S$ conversion [114].

The comparison of catalytic activity and selectivity, taking into account the stability of oxides, allowed Ismagilov and co-authors [115] to find that iron oxide is the most effective catalyst for the reaction of oxidation of hydrogen sulfide to elementary sulfur. Bulk iron oxide catalysts demonstrate high activity at 250 °C, providing almost 100% conversion of hydrogen sulfide at sufficiently high selectivity [112]. It was also shown that the method of iron oxide preparation does not significantly affect the conversion of hydrogen sulfide and the selectivity of the process. The effects of additives of K, Cr, Ag, Ti, V, Mn, and anions: Cl-, $SO_4^{2-}$, $PO_4^{3-}$ in the amount of 1–5% to the initial catalyst on catalytic properties were investigated. In particular, it was demonstrated that the introduction of chromium ions leads to a decrease in the activity, and vanadium ions and $SO_4^{2-}$ to a decrease in the selectivity. A particular feature of the studied catalysts is their ability to oxidize hydrogen sulfide in the presence of hydrocarbons of natural gas without their involvement in the reaction. A significant advantage of these catalysts is their ability to selectively oxidize hydrogen sulfide under an over stoichiometric $O_2/H_2S$ ratio.

An increase in the selectivity of the iron oxide catalyst at elevated temperatures can be achieved by deposition of the active component on an alumina support. The most active catalyst at a temperature of 300 °C is the catalyst of the following composition: 0.5 wt.% $Fe_2O_3/Al_2O_3$ [116]. 34A mixed catalytic system was investigated as a catalyst for partial oxidation of hydrogen sulfide, which consists of iron and titanium oxides (anatase modification). It is proposed to use this system in the process of two-stage oxidation of hydrogen sulfide (at the concentrations of hydrogen sulfide over 5 vol.%). When conducting the process in a temperature range of 200–300 °C, it is possible to remove hydrogen sulfide in the form of elementary sulfur with an efficiency close to 98–99% [117].

It is necessary to emphasize the work creating a catalyst for the direct oxidation of hydrogen sulfide made in VEG-Gasinstitut and the University of Utrecht (The Netherlands) under the general scientific leadership of Professor J. Geus [118–120]. The overall goal of these works is to create an effective and highly selective catalyst for the direct oxidation of hydrogen sulfide by using alumina with a low specific surface area ($\alpha$-modification). It was experimentally shown (Figure 7) that the selectivity of the iron oxide supported on to $\alpha$-alumina had higher selectivity than the catalyst on $\gamma$-alumina even in conditions of a significant excess of oxygen in the reaction mixture. The authors' explanation of the observed results is as follows. By the use of $\alpha$-alumina, it is possible to synthesize the catalyst in which the active component evenly covers the support surface and prevents the diffusion of reagents to the surface of alumina, which has been shown to actively catalyze the reverse Claus reaction-the interaction of sulfur and water vapor to form hydrogen sulfide and sulfur dioxide. This iron catalyst was specially designed for the created SUPERCLAUS® process.

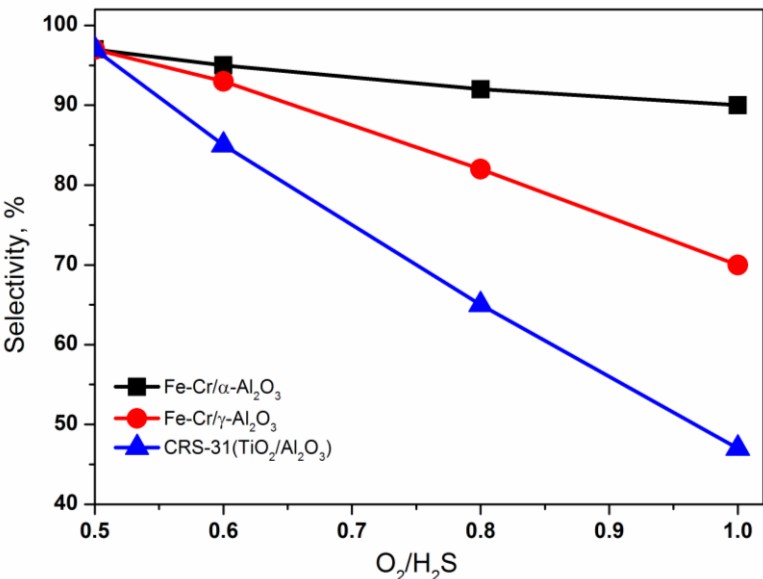

**Figure 7.** The dependence of the selectivity of the process of direct $H_2S$ oxidation on the ratio $O_2/H_2S$ in the initial mixture (adapted from [118]).

The same authors undertook interesting attempts [121] to create effective catalytic systems based on various composite systems and, in particular, alloys of type Hastelloy-X and Inconel with fillers-SUPERCLAUS® commercial catalysts (the amount of additive did not exceed 2 wt.%). The simultaneous use of such materials as catalysts and reactor construction materials was proposed, that is, the effective combination of construction and catalytic properties. It was shown that such systems in the future could make a serious competition to classic reactors with bulk catalysts. Particular emphasis was bestowed on the possibility of using such structures (combined reactor-catalyst) to carry out the hydrogen sulfide oxidation process at a high $H_2S$ content, given the high heat engineering characteristics of the developed materials. However, it was indicated that the upper limit of the content of hydrogen sulfide for the effective operation of these systems should not exceed 10 vol.%.

New possibilities extending the range of the application of the technology of direct $H_2S$ oxidation are provided by monolithic honeycomb catalysts, which possess several technological advantages over granulated catalysts (most important of them low pressure drop), especially for the gases with low excessive pressure and for purposes when the pressure loss is unacceptable.

For the first time, such studies were undertaken by the research team under the guidance of Professor Z.R. Ismagilov. Successful pilot and experimental-industrial tests of the direct oxidation process in reactors with monolithic catalysts of the honeycomb structure to purify the tail gases of the Claus process and geothermal steam are reported [122–124].

Laboratory studies of the catalysts for direct oxidation of hydrogen sulfide in the form of monolithic catalysts of the honeycomb structure were also reported by Italian scientists [125]. Monoliths from cordierite (9 channels) from 10 to 50 mm long, 6 mm wide, and 6 mm high with 226 channels per square inch (CPSI) were used as a substrate for coating. The commercial ceria-zirconia composition (EcoCat) having the initial solids content of 40 wt.% was deposited on cordierite. The active phase ($V_2O_5$) deposition was carried out from an aqueous solution of ammonium metavanadate ($NH_4VO_3$). The authors reported that at a temperature of 200 °C, contact time >200 ms and initial $H_2S$ content 500 ppmv, high conversion of $H_2S$ (90%) and a very low selectivity toward $SO_2$ (3%) were obtained.

Similar data are given by Eom and co-authors [126], where the results of studies on the use of selective catalytic oxidation to remove hydrogen sulfide from landfill gas using

monolithic catalysts of the honeycomb structure are described. The efficiency of removing $H_2S$ at a temperature of 200 °C was the highest for the $V/TiO_2$ catalyst obtained by incipient wetness impregnation. The optimal content of vanadium is 10% by weight. In addition, it was shown that the selectivity to sulfur and minimization of the formation of $SO_2$ substantially depends on the $O_2/H_2S$ ratio. It is shown that with increasing the number of CPSI, the honeycomb catalyst productivity can be significantly increased. The efficiency of $H_2S$ removal also increases with an increase of the specific surface ($m^2/m^3$). The analysis of the long-term operation of a honeycomb catalyst at the cleaning of landfill gas with the composition including $CH_4$ and $CO_2$ (typical components) showed that the purification degree is more than 90%. In addition, the catalyst's performance can be restored by thermal regeneration at sufficiently "soft" conditions (400 °C, 3 h in airflow).

## 19. Description of Modern Industrial Methods Based on the Process of Direct H2S Oxidation

The Catasulf® process of the German company BASF (Ludwigshafen, Germany) [127] is based on the reaction of the oxidation of the acid gas containing 5–15% $H_2S$ (I) in the tubular reactor 1 (Figure 8). The tube space of reactor 1 is filled with a special highly selective catalyst, which is a mixture of aluminum, nickel, and vanadium oxides, and the inter-tube space is cooled with a high-boiling liquid silicon coolant (II), which, circulating, transfers the removed heat to the refrigerator 2. Gases emerging from reactor 1, (III) are cooled in the sulfur condenser 3 and fed into the adiabatic reactor 4, where there is a further interaction of hydrogen sulfide with sulfur dioxide. The resulting sulfur is separated in the second consecutive condenser 5. Removing the sulfur in the first stage is 94%, after the adiabatic reactor up to 97.5%. It is supposed that by increasing the number of stages, it is possible to attain the degree of sulfur extraction of 99.99%. At the oil refining plant, Ludwigshafen (Germany), the only Catasulf® industrial installation is operated.

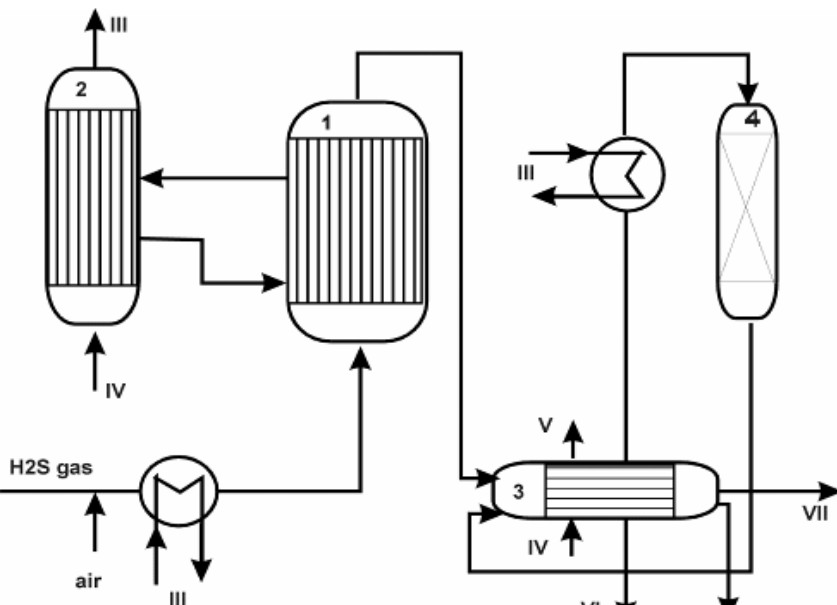

**Figure 8.** Schematic of Catasulf® process Catasulf®, perhaps, the only "active" large-capacity technology to obtain sulfur based on the reaction of the direct oxidation of hydrogen sulfide (adapted from [127]).

The Sulfatreat® DO process of the company M-I Swaco [128,129] is a technology for the purification of associated oil gases from hydrogen sulfide by direct oxidation. The process is carried out at a temperature of 175 °C, a pressure of up to 65 bar and allows the treatment of gases containing up to 3 vol.% hydrogen sulfide. The catalyst is a mixture of metal oxides of transitional valence, promoted with alkali metals oxides. The results of

tests of the experimental installation showed that the H2S concentration in the purified gas does not exceed 950 ppmv. The degree of sulfur removal, in this case, was more than 88%, with only a slight conversion of the hydrocarbon part.

Other industrial processes based on this reaction are designed exclusively for cleaning tail gases of existing sulfur-producing installations. One of the most common methods is BSR/Selectox® of Unocal and Ralph M. Parsons companies. In this process, the exhaust gases of the Claus installation are reduced in a catalytic reactor by synthesis-gas formed in a special generator by a steam reforming of natural gas. Then the resulting gas is cooled, mixed with air, and hydrogen sulfide is subjected to selective oxidation to sulfur at a temperature of 200–230 °C. The use of a special vanadium oxide catalyst Selectox-67 allows attaining the selectivity of oxidation to sulfur of ca. 100%. The Beavon-Selectox process ensures the degree of sulfur removal of 98.5–99.5% at a relatively low installation cost (about 50–60% of the cost of a Claus installation [130]. The first such installation was launched in 1978 in Germany.

Somewhat later, other technologies similar to the Beavon-Selectox process were developed. Among them are the most famous MODOP® of Mobil Oil Corp (Dallas, TX, USA) [131]. and SUPERCLAUS® of Comprimo BV Company (Dallas, TX, USA) [42]. They differ from the Beavon-Selectox process by using other catalysts (CRS-31 for MODOP and a special highly selective iron oxide catalyst for the SUPERCLAUS® process), and by the fact that the oxidation of hydrogen sulfide is carried out in two stages. In addition, in the SUPERCLAUS® process, it is possible to use the H2S direct oxidation stage without the hydrogenation stage due since in the Claus installation, the process is carried out with an excess of hydrogen sulfide (this allows, among other things, protect the catalyst in the Claus reactors from sulfation). These processes provide the total degree of sulfur removal of 99.3–99.5%. The first two MODOP® (Dallas, TX, USA) installations were put into operation in Germany in 1983 and 1987 and two SUPERCLAUS® installations in 1988 in Germany and 1989 in Holland [132–134].

The advantage of the processes described above is the possibility to supply air for the oxidation of hydrogen sulfide in a small excess compared to stoichiometry, which simplifies the control of the process in the conditions of variation of the composition and flow rate of the reaction mixture.

However, the use of direct heterogeneous catalytic oxidation of hydrogen sulfide is significantly limited because of intense heating of a fixed catalyst bed due to high heat generation. At the Boreskov Institute of Catalysis SB RAS, the technology of direct oxidation of hydrogen sulfide in a reactor with a fluidized catalyst bed was developed, which is largely free of these shortcomings.

A research program was implemented under which the effects of temperature and concentration of components on the kinetic parameters of the direct hydrogen sulfide oxidation process were studied. The oxidation of hydrogen sulfide in the composition of hydrocarbon-containing mixtures, the kinetic parameters of the hydrogen sulfide process for various catalytic systems, and the elementary stages of the process were investigated, and the activities of a wide range of supported oxide catalysts in the target reaction were measured.

The main results of the research are the following:

A wide range of supported catalysts meeting the requirements for catalytic systems operating in the reactor with a fluidized bed by their structural and mechanical characteristics (i.e., high mechanical strength and thermal stability) were synthesized and characterized [113,135,136]. Honeycomb catalysts have also been developed for the H2S oxidation process [137–139].

When studying the regularities of the reaction on the magnesium-chromium oxide catalyst, it was found that the following equation could describe by reaction kinetics [Equation (24)] [140]:

$$W = \kappa \cdot (C_{H_2S})^m \cdot (C_{O_2})^n \tag{24}$$

The orders of *m* and *n* have similar values close to 0.5.

This value of the observed order of the hydrogen sulfide oxidation reaction indirectly indicates that the first elementary stage of the process is the dissociative adsorption of hydrogen sulfide on the catalyst's surface. The activation energy is significantly lower than the value found for alumina, and it is about 8.1 kcal/mol [140].

The effect of hydrocarbons in the composition of the gas mixture on the parameters of the reaction of the direct oxidation of hydrogen sulfide was studied, which is a fundamental issue in developing the scientific foundations for the purification of hydrogen sulfide containing fossil fuels.

As can be seen from the results presented in Figure 9, the temperature areas of the effective action of catalysts for the selected reactions are sufficiently separated, that is, in the temperature range of 220–260 °C, where the sulfur yield achieved is close to 100% propane oxidation reaction proceeds at a low rate [141].

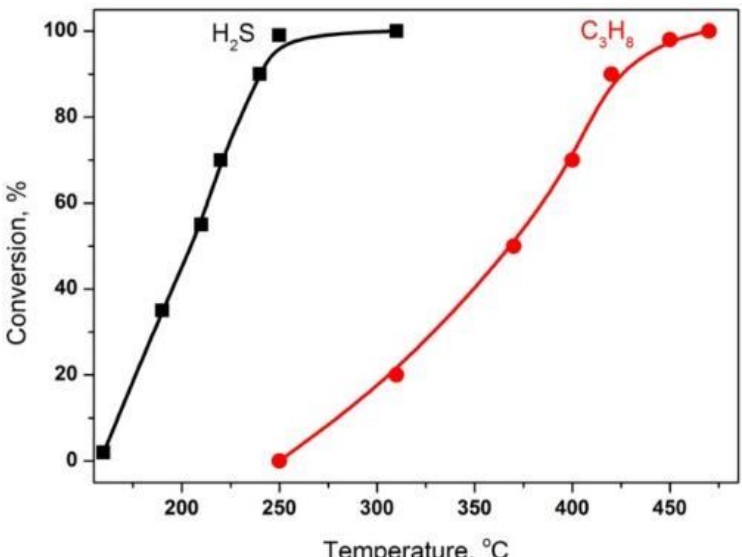

**Figure 9.** Results of laboratory experiments on the separate oxidation of propane and hydrogen sulfide over a MgCr$_2$O$_4$/$\gamma$-Al$_2$O$_3$ catalyst (Catalyst: MgCr$_2$O$_4$/$\gamma$-Al$_2$O$_3$; residence time: 0.8 s; C$_{H2S}$: 30 vol.%; C$_{C3H8}$: 15 vol.%).

Since kinetic data is quite formal and does not give unequivocal information about the mechanisms involved in the process, attempts have been made to study the elementary reaction stages using spectral methods [142–144]. To this end, three systems were selected:

- Baseline magnesium-chromium oxide catalyst MgCr$_2$O$_4$/$\gamma$-Al$_2$O$_3$
- Iron oxide catalyst Fe$_2$O$_3$/$\gamma$-Al$_2$O$_3$
- $\gamma$-alumina $\gamma$-Al$_2$O$_3$.

FTIR spectroscopy of the adsorbed CO revealed that all the catalysts had both Lewis and Broensted acid sites on the surface (Figure 10). However, the nature, strength, and number of sites varied according to the type of catalyst. H$_2$S adsorption and the formation of intermediates occurred on Lewis acid sites, as confirmed by the disappearance of LAS bands after H$_2$S adsorption.

The adsorption of H$_2$S on the surface of catalysts (Figure 11) led to the formation of two types of surface species: sulfates (I) at 1100 cm$^{-1}$ and (II) registered at higher frequencies 1264 and 1342 cm$^{-1}$, corresponding to organic sulfates. The sulfates of type (I) formed on $\gamma$-alumina at 100 °C, while type (II) formed at 250 °C. The formation of these two species was detected at much lower temperatures on Fe$_2$O$_3$/$\gamma$-Al$_2$O$_3$ and MgCr$_2$O$_4$/$\gamma$-Al$_2$O$_3$ due to their higher oxidative activity.

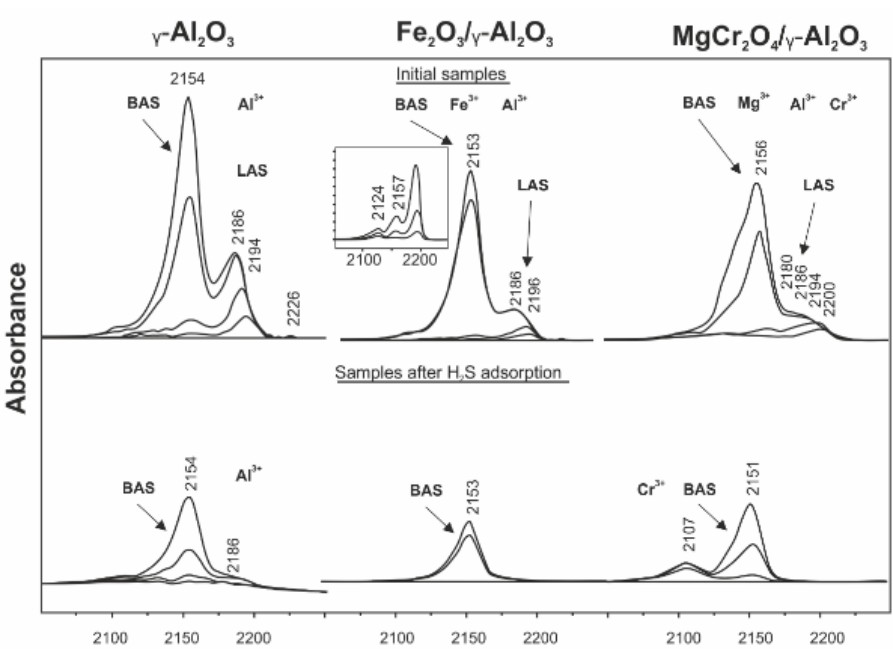

**Figure 10.** IR spectra of adsorbed CO: initially and after H2S adsorption.

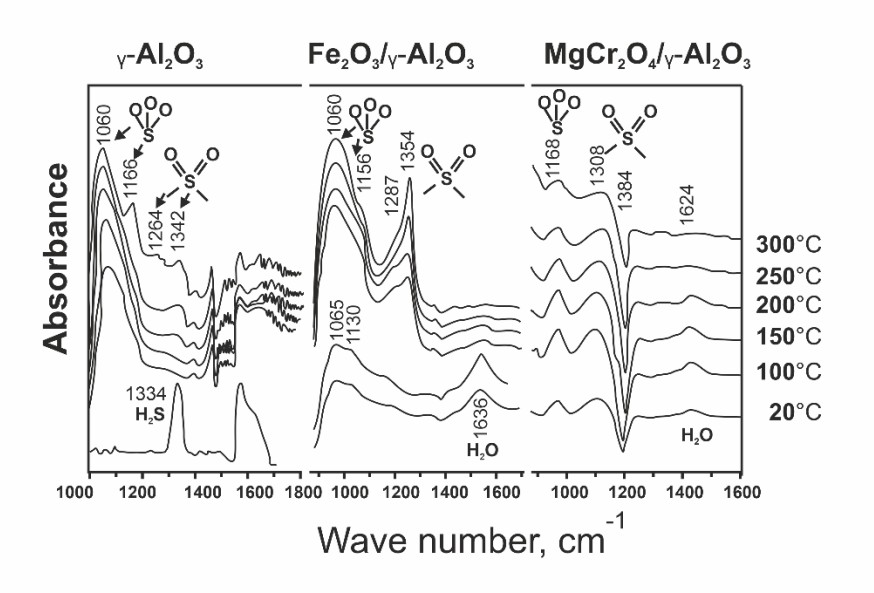

**Figure 11.** IR spectra of samples of different catalysts after adsorption of 20 torr of H2S at various temperatures.

The DRS study (Figure 12) revealed that various types of elemental sulfur, i.e., $S_4$-$S_8$, formed on the catalyst surface during the reaction depending on the nature of the catalyst.

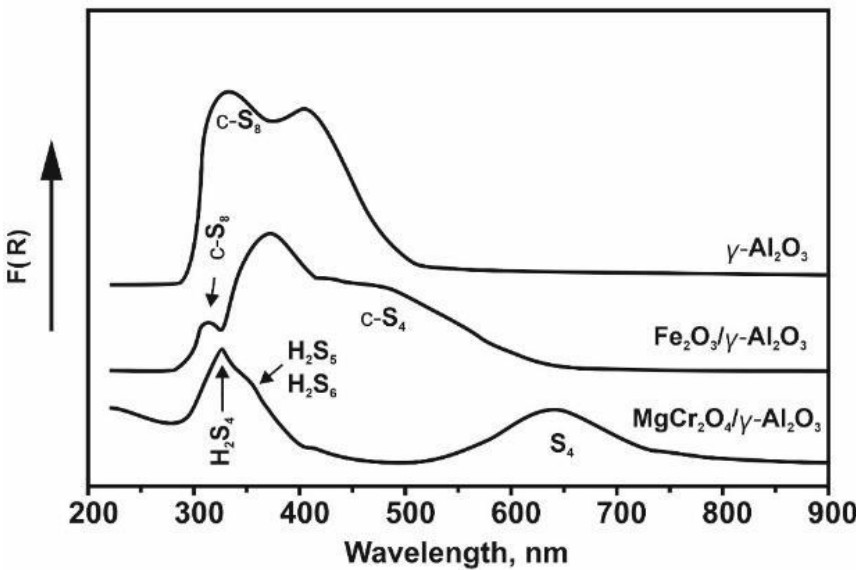

**Figure 12.** Different sulfur species formed on the surface of catalysts after H2S oxidation, as detected by UV DRS (30 torr H2S, 1 h, T = 250 °C).

Based on the data obtained, the reaction mechanism for the direct oxidation of hydrogen sulfide can be represented by the schematic depicted in Figure 13.

In the first stage, the hydrogen sulfide is adsorbed on the surface of the catalyst. The adsorption can occur (i) through the participation of the LACs and the sulfur atom of the hydrogen sulfide molecule, (ii) through the participation of the BACs and the sulfur atom with the formation of hydrogen bonds, or (iii) through the participation of a particular catalyst center, e.g., surface oxygen and the proton of the H2S molecule. The adsorption on Lewis acid centers leads to the greatest activation of the hydrogen sulfide molecule.

Next, the hydrogen sulfide molecule adsorbed on the Lewis acid center can interact with a neighboring oxygen atom of the catalyst or a hydroxyl group. This process can lead to the dissociation of hydrogen sulfide molecules to form a hydroxyl group or water.

The oxygen of the catalyst surface oxidizes the formed surface particles to form surface SO2 groups, which, upon interacting with the hydrogen sulfide molecule from the gas phase or with an adsorbed hydrogen sulfide molecule, will yield the final reaction products, i.e., elementary sulfur and water, via the surface Claus reaction.

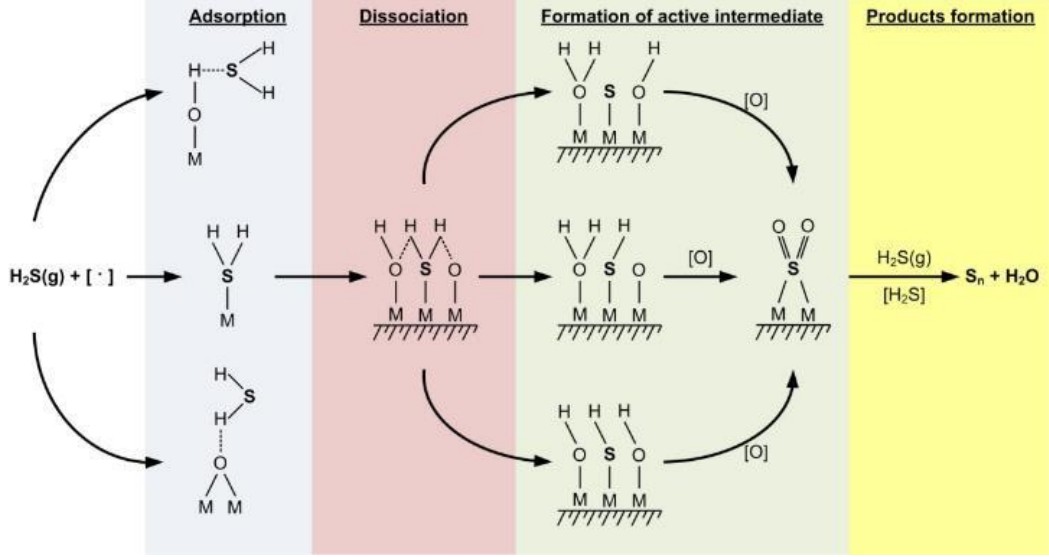

**Figure 13.** Proposed mechanism of H2S oxidation on oxide catalysts.

A raw hydrogen sulfide-containing gas is supplied to the reactor with a fluidized catalyst. Simultaneously, oxygen (or air) is fed into the catalyst bed via a separate flow. Before the gas stream supply, the catalyst bed is heated to initiate the catalytic reaction. The excessive heat of the exothermic reaction of $H_2S$ oxidation is efficiently removed by a heat-exchanger in the fluidized bed. The bed temperature is maintained within the preset range (280–320 °C) with high uniformity by regulating the amount of heat removed from the bed with a heat-exchange agent.

The technology was successfully tested on a pilot and industrial scale in Russia's largest sour gas fields, refineries, and gas processing plants.

### 20. Developments of the Boreskov Institute of Catalysis SB RAS Regarding the Creation of Processes of Heterogeneous Catalytic Oxidation of Hydrogen Sulfide for the Treatment of Various Gases

At the Boreskov Institute of Catalysis SB RAS, various technologies for direct catalytic oxidation of hydrogen sulfide have been developed.

Using data from the development of highly exothermic catalytic processes, in particular, processes of combustion of organic fuels in fluidized catalyst bed reactors [145–148], a new technology using highly concentrated hydrogen sulfide-containing gas was proposed, the essence of which consists of a reaction conducted in a fluidized bed catalytic reactor-Modification 1 (Figure 14).

Due to moderate temperatures (250–320 °C) applied in this technology, no hydrocarbon cracking reactions were observed. Thus, hydrogen formation seems unlikely. This assumption was confirmed by the results of the pilot and industrial tests, that showed:

1.  The preservation of qualitative and quantitative composition of hydrocarbons, and
2.  The absence of hydrogen in the reaction products after the reactor.

The primary source, which is potentially dangerous from the viewpoint of explosion safety, is the catalytic reactor in which the formation of hydrogen sulfide mixtures at explosive concentrations (i.e., 4.3–45.5 vol.% in the air) is possible. However, this problem is minimized by the following factors:

1.  Hydrogen sulfide is almost completely removed on the first three dm of the catalyst bed; that is, its concentration drops significantly, i.e., to below the explosion limit.
2.  The reaction proceeds solely on the surface of the catalyst, so the transition of the process into the reactor volume proceeding according to the homogeneous chain (explosive) mechanism is excluded. Thus, the catalyst bed acts essentially as an effective flame arrester.

A feed of hydrogen sulfide-containing gas is supplied to the reactor via a fluidized catalyst bed. Simultaneously, oxygen (or air) is fed into the catalyst bed via a separate flow. Before the gas stream supply, the catalyst bed is heated to initiate the catalytic reaction. The excessive heat of the exothermic reaction of $H_2S$ oxidation is efficiently removed by a heat-exchanger in the fluidized bed. The bed temperature is maintained within the preset range (280–320 °C) with high uniformity by regulating the amount of heat removed from the bed with a heat-exchange agent [49,149–153].

At the same time, there is a treatment problem, i.e., a low pressure drop is required in the reactor, for gases with low concentrations of hydrogen sulfide, such as the tail and ventilation gases of various chemical industries, as well as the purification of energy carriers, such as oil-associated gases and geothermal steam where the pressure loss is extremely undesirable.

To solve these problems, the reaction was conducted in the reactor with a monolithic catalyst with a honeycomb structure (process modification 2, Figure 15). Such catalysts have some advantages, in particular a low pressure drop and a high ratio of the outer surface area to volume [154]. Technologies have been tested on a pilot and experimental industrial scale, and their abilities to clean various gases containing hydrogen sulfide have been demonstrated (Tables 2 and 3, Figures 16–20).

**Table 2.** Pilot and experimental industrial tests of the technology of direct catalytic oxidation of hydrogen sulfide (Modification 1-fluidized bed) [140].

| # | Location Object $H_2S$ Content | Operation Conditions | | Year | $H_2S$, % |
|---|---|---|---|---|---|
| | | Scale | Gas Supply | | |
| 1 | Astrakhan sour gas field Natural gas $C_{(H2S)} = 27$ vol.% | Pilot | up to 50 $nm^3/h$ | 1987 | 98 |
| 2 | Astrakhan sour gas field Natural gas $C_{(H2S)} = 27$ vol.% | Pilot | up to 50 $nm^3/h$ | 1988 | 98 |
| A3 | Astrakhan sour gas field Natural gas $C_{(H2S)} = 27$ vol.% | Pilot | up to 20 $nm^3/h$ | 1991 | 98 |
| 4 | Ufa Refinery Hydrodesulfurization gas $C_{(H2S)} = 70$ vol.% | Pilot | up to 50 $nm^3/h$ | 1990 | 98 |
| 5 | Shkapovo GPP Acid gas from amine unit $C_{(H2S)} = 65$ vol.% | Semi-industrial | up to 350 $nm^3/h$ | 1995 | 98 |
| 6 | Bavly oil field Acid gas from amine unit $C_{(H2S)} = 65$ vol.% | Semi-industrial | up to 70 $nm^3/h$ of acid gas | 2004–2009 | 99.5 |

**Table 3.** Pilot and experimental industrial tests of the technology of direct catalytic oxidation of hydrogen sulfide (Modification 2-Honeycomb Catalyst) [140].

| # | Location Object $H_2S$ Content | Operation Conditions | | Year | $H_2S$, % |
|---|---|---|---|---|---|
| | | Scale | Gas Supply | | |
| 1 | Novo-Ufimsky Refinery Tail gas of Claus process $C_{(H2S)} = 2$ vol.% | Pilot | up to 20 $nm^3/h$ | 1989-1990 | 98 |
| 2 | Astrakhan GPP Tail gas of Claus process $C_{(H2S)} = 2$ vol.% | Pilot | up to 20 $nm^3/h$ | 1991 | 98 |
| 3 | Orenburg GPP Gases of zeolites regeneration $C_{(H2S)} = 2$ vol.% $C_{(RSH)} = 5$ vol.% | Pilot | up to 20 $nm^3/h$ P up to 0.5 MPa | 1990 | 98 |
| 4 | Kamchatka peninsula Geothermal steam $C_{(H2S)} < 1$ vol.% $C_{(H2O)} > 99$ vol.% | Fixed bed Pilot | up to 0.5 tn. steam/h P up to 1.0 MPa | 1989-1990 | 99.9 2500 h of continuous operation |
| 5 | Novo-Ufimsky Refinery Tail gas of Claus process $C_{(H2S)} = 2$ vol.% | Semi-industrial | up to 7000 $nm^3/h$ | 1994 | 98 |

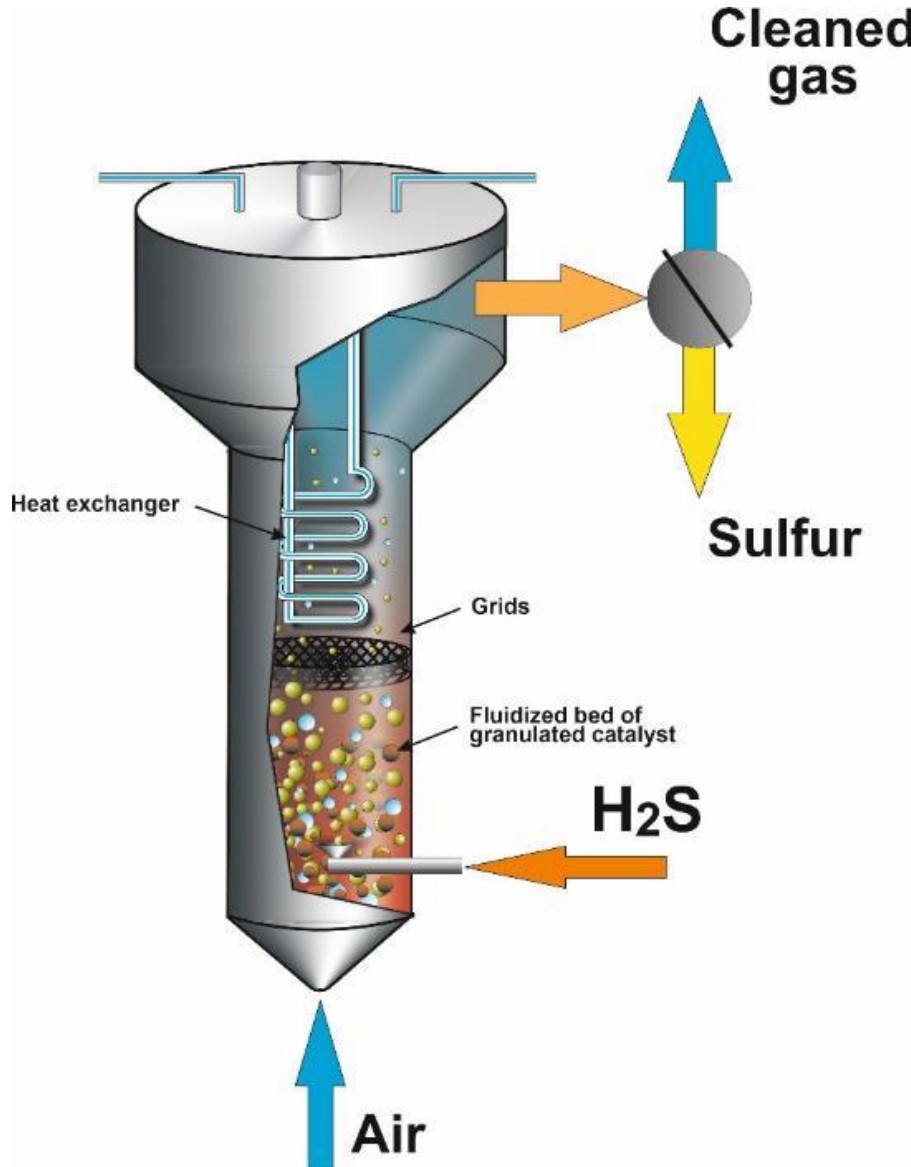

**Figure 14.** Direct catalytic oxidation in a reactor with a fluidized catalyst bed. Basic engineering concept.

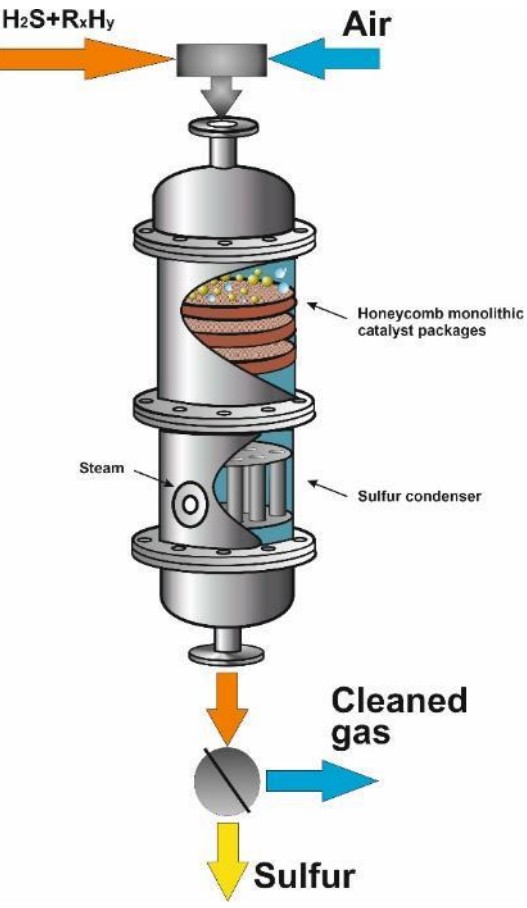

**Figure 15.** Direct catalytic H2S oxidation in a reactor via a monolithic catalyst with a honeycomb structure.

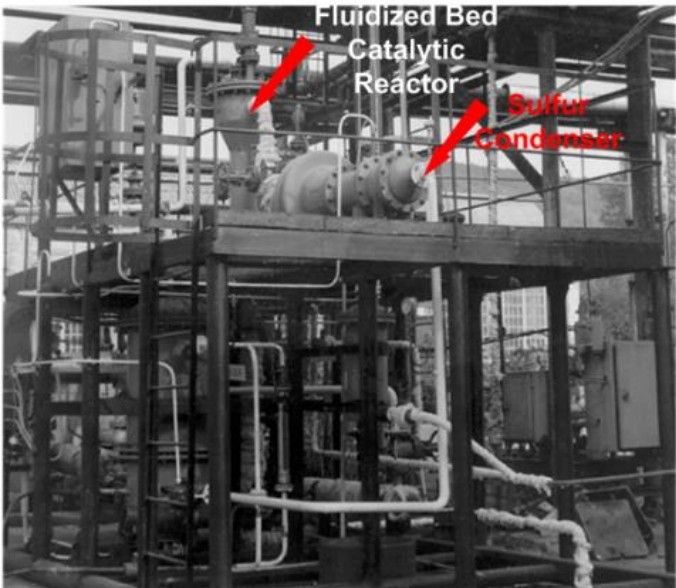

**Figure 16.** Pilot Plant at the Ufa Refinery.

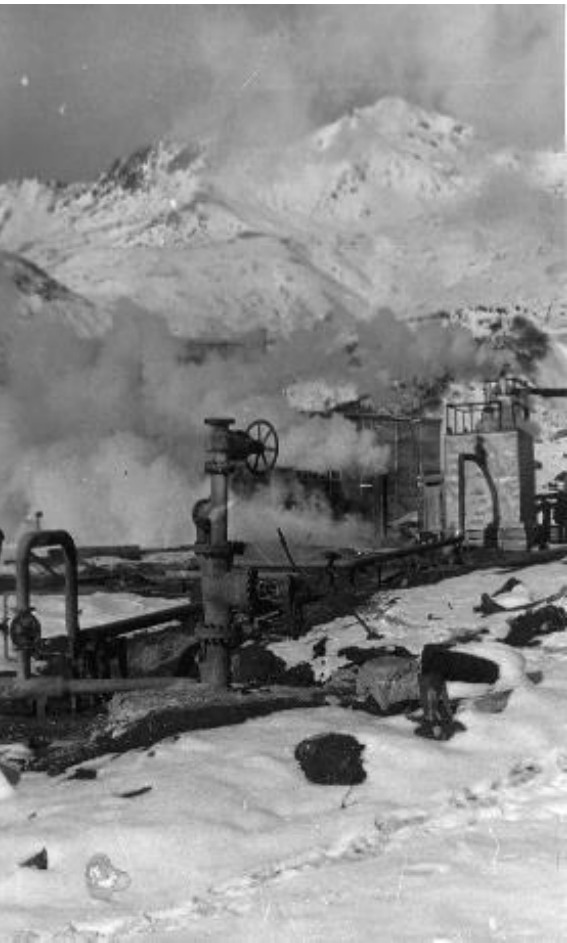

**Figure 17.** Mutnovskoe deposit of geothermal steam.

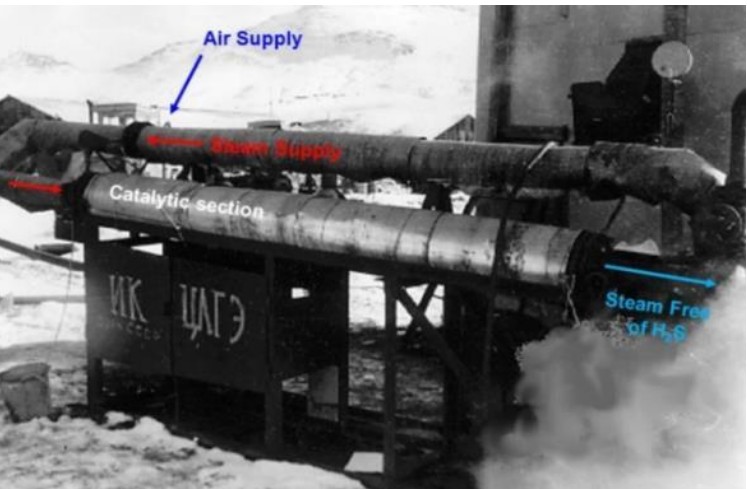

**Figure 18.** Pilot plant for H2S removal from geothermal steam.

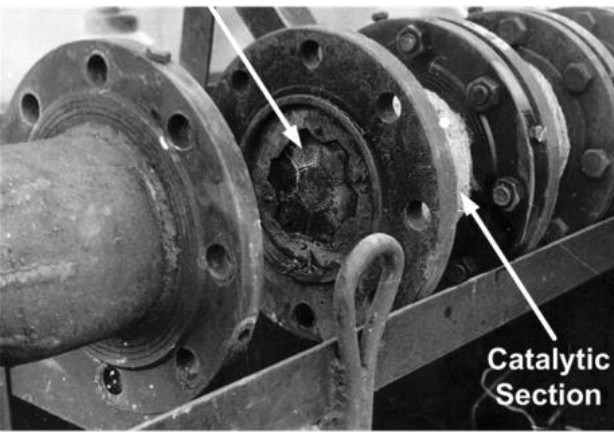

**Figure 19.** Catalytic segment after 2500 h of continuous operation.

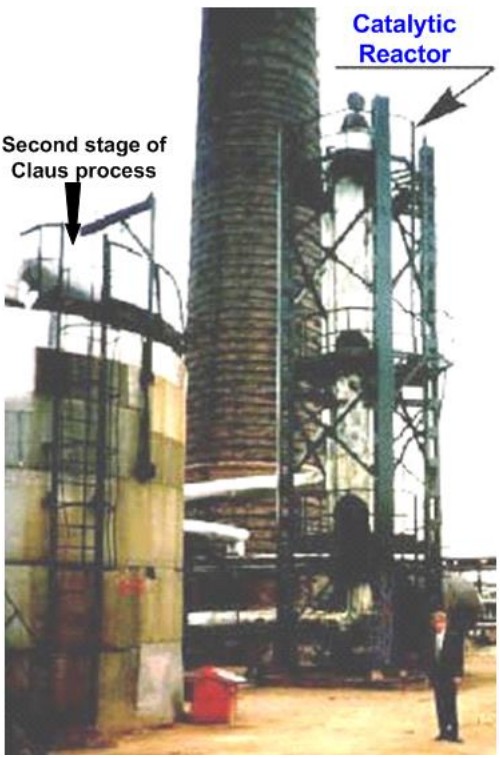

**Figure 20.** Semi-industrial installation for the direct oxidation of hydrogen sulfide via monolithic catalyst with a honeycomb structure. Tail-gas of the Claus process.

## 21. Installation for H2S Recovery From Acid Gas after the Amine Treatment of Oil-Associated Gases at Bavly Gas Shop of PJSC Tatneft

The use of associated petroleum gas (APG) is strictly regulated according to the legislation implemented by the Russian government on 8 November, 2012 (#1143, edited on 17 December, 2016), which states that "Regarding peculiarities of the cost calculation for the negative environmental impact during emission of pollutants generated upon combustion using flare facilities and/or associated petroleum gas scattering". The legislation also includes a statement on the peculiarities of cost calculation for a negative environmental impact due to the emission of pollutants generated by facilities using a combustion flare and/or associated petroleum gas scattering.

Additionally, APG is a source of the propane-butane fraction for petroleum chemistry companies in Russia. This fraction is often in short supply. In order to address the issue of the primary removal, sorption facilities for amine treatment have been developed to remove hydrogen sulfide and transport hydrocarbon components to the appropriate sites of further treatment. However, the problem is addressed only partially, as the hydrogen sulfide released is burnt with flares.

Typical examples of the implementation of such an approach are the Bavlinsky gas workshop, PJSC Tatneft, Shkapovskiy, and Tuymazinskiy gas processing plants of PJSC ANK Bashneft.

In 2011, an industrial installation with a fluidized catalyst bed for the removal of hydrogen sulfide from acid gases from the amine treatment of oil-associated gases was created and put into operation by the Boreskov Institute of Catalysis SB RAS [155–158] at the PJSC Tatneft Bavly gas shop (Figures 21–23).

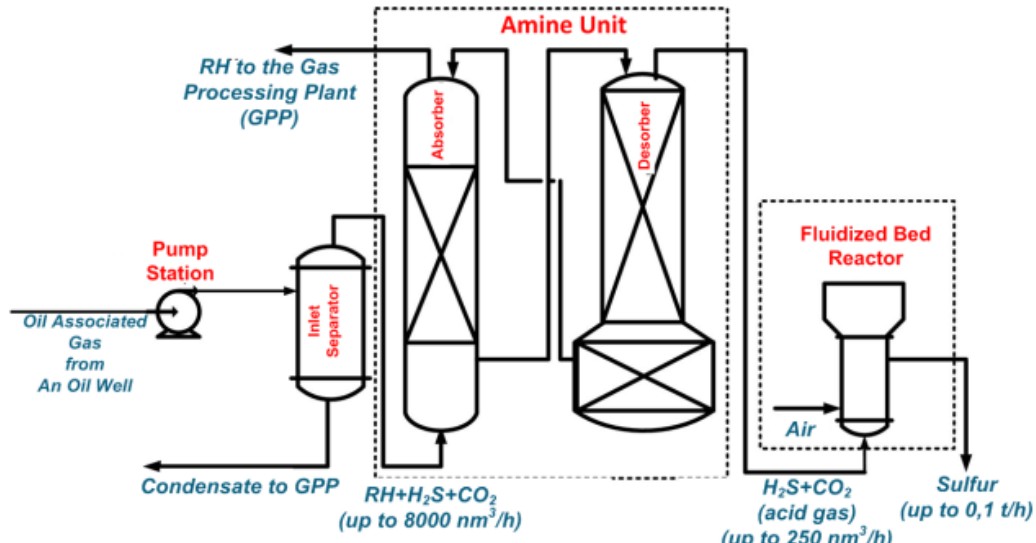

**Figure 21.** Bavly gas shop of PJSC Tatneft. Purification of associated oil gas. Amine treatment and direct oxidation.

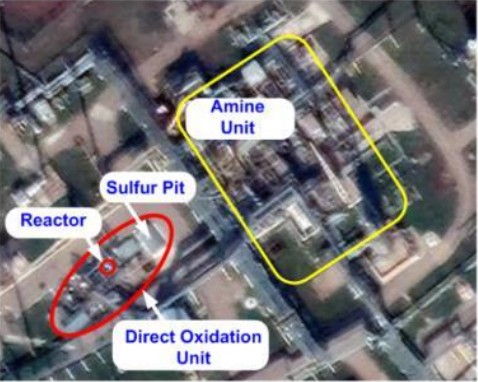

**Figure 22.** Bavly gas shop of PSC Tatneft. Purification of oil associated gas. Amine treatment and direct oxidation. Source: Satellite photo.

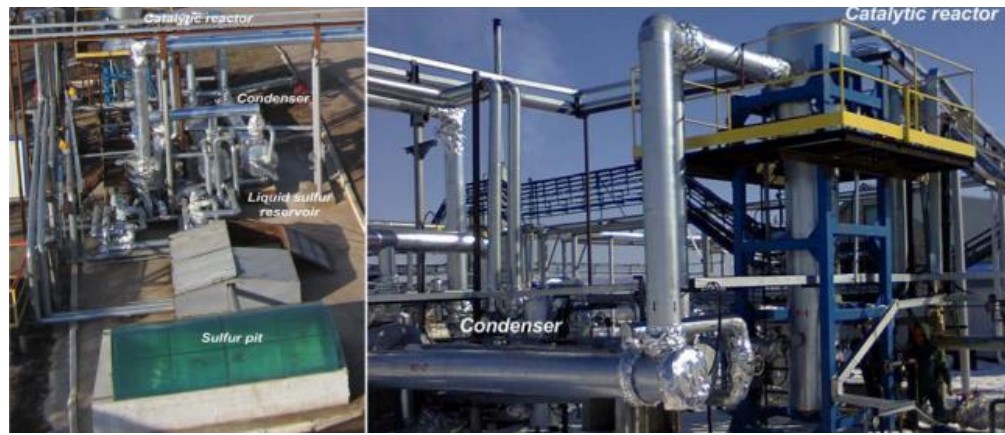

**Figure 23.** Industrial unit with a capacity of acid gas of up to 250 nm$^3$/h, which has been in continuous operation since 2011. The hydrogen sulfide content is 30–65 vol.%.

The main feature of the initial feed is the extreme instability of the input gas parameters; see Figure 24.

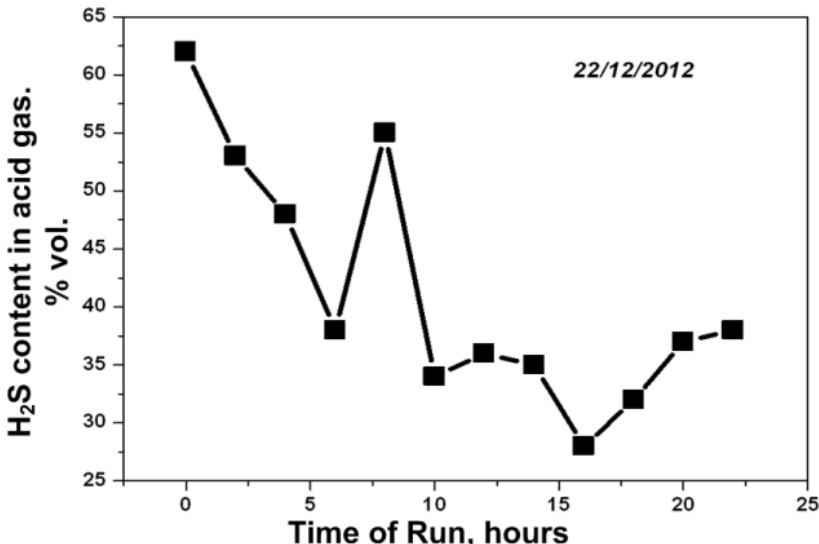

**Figure 24.** Fluctuations of H$_2$S content in the feed gas subjected to purification.

However, the developed computer control system made it possible to rapidly adjust the air and coolant flows to maintain the preset temperature in the catalytic reactor.

The quality of the resulting sulfur (Figure 25) surpassed the Russian National Standard #127.1-93 (commercial grade sulfur 9990).

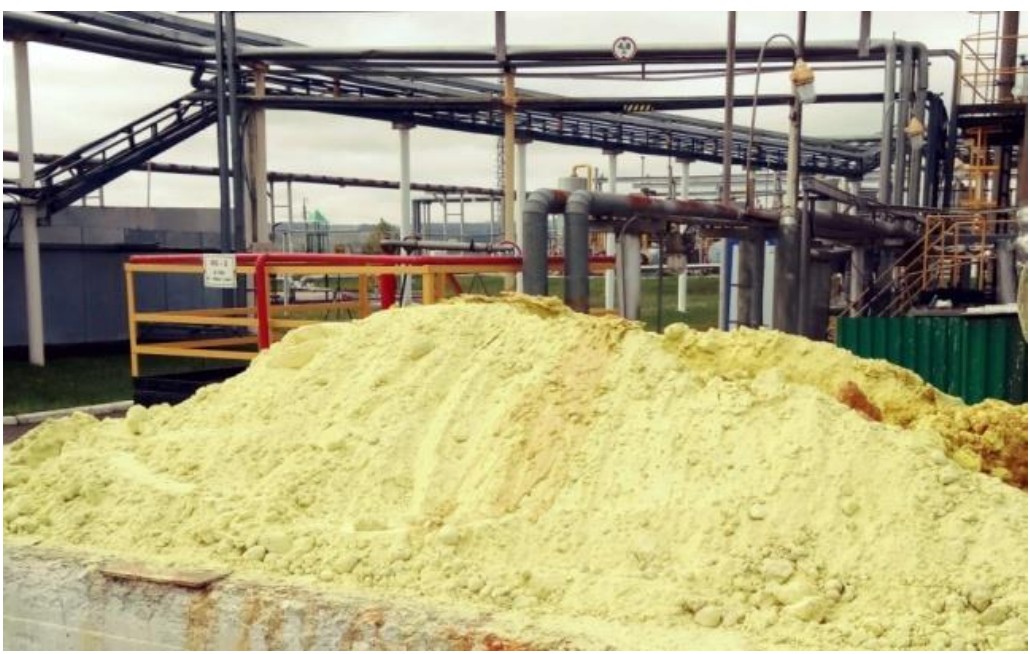

**Figure 25.** Sulfur produced at the Bavly gas shop of PJSC Tatneft.

The main results of the operation of the installation in the Bavly gas shop are given below:

- Over 1 billion m³ of purified gas produced
- 6000 tons of hydrogen sulfide converted to elementary sulfur
- Emission of 12,000 tons of sulfur dioxide and sulfuric acid (340 railway tanks) into the atmosphere prevented;
- Environmental damage amounting to about 2.9 billion rubles avoided
- One-stage technology with computer control providing stable operation with variable parameters in terms of the acid gas (for example, hydrogen sulfide content).

## 22. Facility for the Purification of Gases Caused by Blowing-off Sour Crude Oil

Strict limitations for the hydrogen sulfide content in oil for pipeline transport have been in place in Russia since 2002. Herewith, the mass fraction of hydrogen sulfide is limited to within 20–100 ppmv (GOST P 51858-2002. Oil. General technical conditions). When purifying 200 g of oil an hour, about 0.1 t of $H_2S$, or, on a yearly basis, 800 t of $H_2S$, are generated. This is particularly relevant because of the short period of transition (2019–2020) regarding technical regulations put in place by the Eurasian Economic Union "Regarding the safety of oil prepared for transportation or use" (TR EAES 045/2017), limiting hydrogen sulfide levels to 20 ppmv. The blowing-off process of $H_2S$ with purified gas (mainline natural gas) is used for oils from the fields in the Volga Ural oil and gas province (Nurlatskoye, Aznakaevskoye, and Aznakaevskoye) with hydrogen sulfide levels up to 600 ppmv.

In this case, a $H_2S$-enriched hydrocarbon flow is formed which then undergoes amine treatment. Meanwhile, the concentrated hydrogen sulfide should be disposed of using the most reasonable method. To this end, the Boreskov Institute of Catalysis SB RAS, in collaboration with specialists from JSC SHESHMAOIL and JSC VNIIUS, constructed an industrial unit (Table 4).

The unit is scheduled to be used on an industrial basis in 2021 [159].

**Table 4.** Main characteristics of the installation for the removal of H$_2$S from gases originating from the blowing off of sour crude oil, designed for JSC SHESHMAOIL.

| # | Parameters | Value |
|---|---|---|
| 1 | Acid gas flow rate after amine unite to the direct oxidation unit, nm$^3$/hour | to 110 |
| 2 | H$_2$S concentration in acid gas, vol.% | 75–90 |
| 3 | Diameter of the fluidized bed reactor, m | 0.52 |
| 4 | Catalyst loading, kg | 185 |
| 5 | Sulfur yield, tons/hour | 0.13 |

Minirefineries and GPPs or plants with a capacity of recycled hydrocarbon sulfurous raw materials of up to 3 million tons of oil per year for refineries and up to 80 million m$^3$ gas per year for GPPs are worthy of special consideration. Such enterprises are becoming rather numerous in the CIS countries. Their main goal is to localize the production of high-quality motor fuels in regions which are distant from large oil and gas processing centers. The low capacity of such production does not allow the creation of full-size hydrogen sulfide utilization units based on the Claus process, and the hydrogen sulfide formed as a result of the primary processing processes is usually burned off.

To solve this issue, at the JSC Condensate (Republic of Kazakhstan), an installation for hydrogen sulfide removal with sulfur production was built. Investors recognized the compact direct oxidation plant as the most rational way to solve the problem from a technical and economical viewpoint.

The installation has successfully passed commissioning and is ready to begin permanent operations.

At present, a plant which will use the hydrogen sulfide formed in the hydrocracking process is being created at the Ust-Luga Complex of PJSC NOVATEK. The technology was selected as a result of a vote, as it proved to be superior to those proposed by other licensers. The acid gas flow rate to the direct oxidation unit after the amine unit is about 170 nm$^3$/h.

The present status of the technology is as follows:

- The basic design of the technology has been finalized.
- The design and working documentation have been presented.
- The various apparatus units have been fabricated (Figure 26);
- The block of the plant has been delivered to the customer (Figure 27);
- The technology achieves the direct catalytic oxidation of hydrogen sulfide via the use of acid gases. It is an alternative to the Claus process (MTU-0.5 Mini Plant, Republic of Kazakhstan).

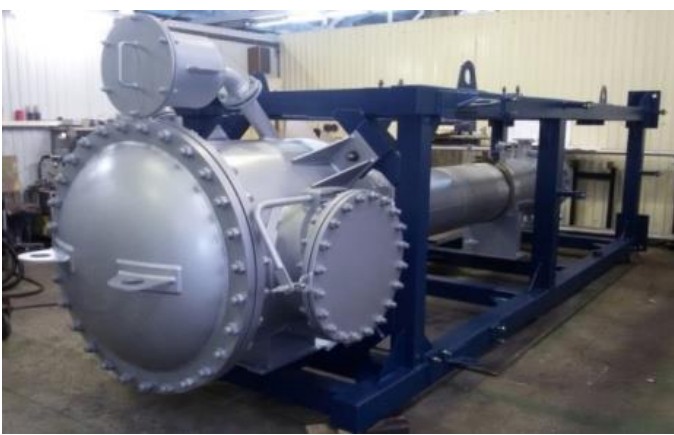

**Figure 26.** Reactor Block.

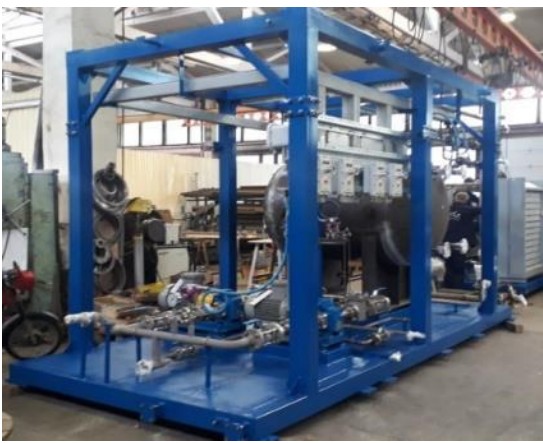

**Figure 27.** Cooling Block.

## 23. Unit for the Direct Oxidation of Hydrogen Sulfide as a Component of the Associated Petroleum Gas

Another application for sour associated gas is as low-debit flows with a capacity of 1000 nm³/h. On the one hand, these flows are environmental pollution sources and may be used with a compact method of purification for the autonomic regeneration of heat energy and electric power for travel heaters, the power supply for gas turbine units, etc.

SMP specialists Neftegaz JSC, BIC, in collaboration with TatNIINentefemash JSC and VNIIUS JSC, developed a production unit to selectively remove hydrogen sulfide directly from APG (Figures 28 and 29) [160–162]. The unit has undergone a complete cycle of industrial tests and is ready for industrial application.

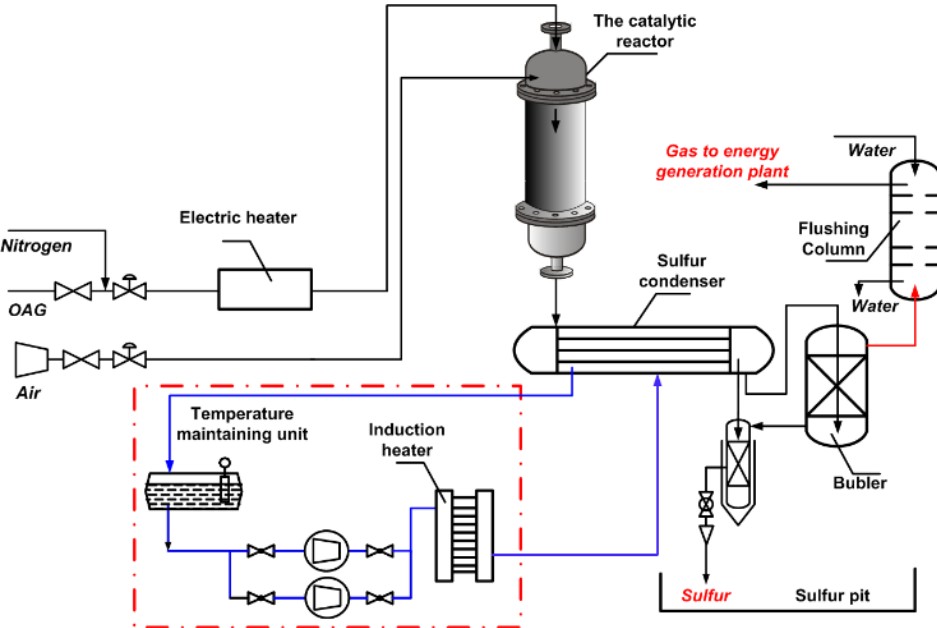

**Figure 28.** Flow-sheet diagram of the purification plant. Direct catalytic oxidation of hydrogen sulfide.

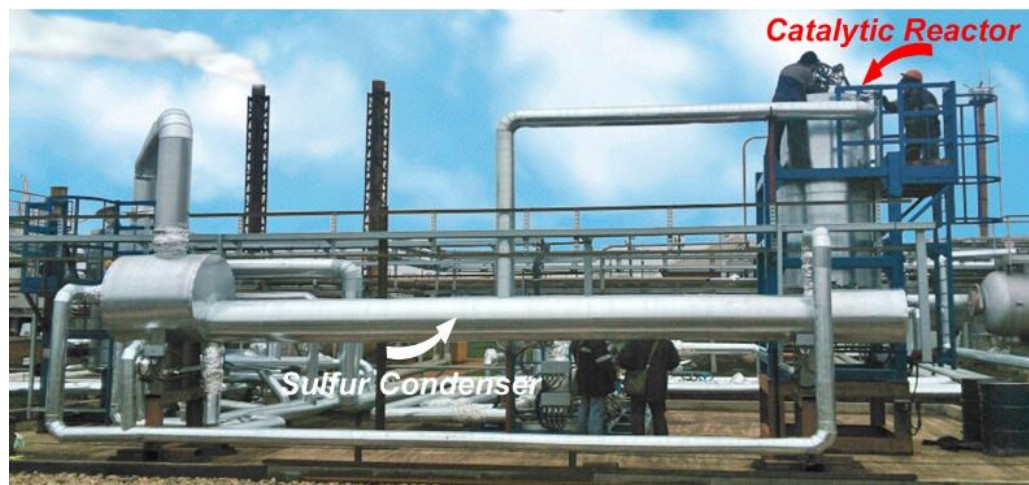

**Figure 29.** Industrial APG purification plant by direct catalytic oxidation.

The most important indicator of the process is its selectivity with respect to the hydrocarbon part of the purified gas. In this regard, a technique was developed to study the composition of the hydrocarbon part of the gas which is able to precisely identify individual components based on GC analysis.

The results are shown in Table 5.

As shown in the given data, hydrocarbon components are preserved during gas purification, and the purified gas can be used to generate thermal and electrical energy with minimal damage to the environment.

**Table 5.** GC analysis of initial and purified gas.

| # | Compound | Initial Feedstock Gas, %Vol. | Purified Gas, %Vol. |
|---|---|---|---|
| 1 | $H_2S$ | 1.50 | <50 ppmv |
| 2 | Water | 0.69 | 2.030 |
| 3 | He | 0.05 | 0.04 |
| 4 | Hydrogen | 0.006 | 0.004 |
| 5 | Oxygen | 0.04 | 0.92 |
| 6 | $CO_2$ | 4.70 | 4.56 |
| 7 | Nitrogen | 39.82 | 41.00 |
| 8 | Ethane | 9.60 | 9.60 |
| 9 | Methane | 25.60 | 24.22 |
| 10 | Propane | 9.96 | 9.80 |
| 11 | iso-Butane | 2.02 | 1.96 |
| 12 | n-Butane | 3.45 | 3,34 |
| 13 | neo-Pentane | 0.003 | 0.003 |
| 14 | iso-Pentane | 1.23 | 1.19 |
| 15 | n-Pentane | 0.85 | 0.81 |
| 16 | Hexanes | 0.32 | 0.31 |
| 17 | Heptanes | 0.07 | 0.07 |
| 18 | Octanes | 0.10 | 0.09 |

The preliminary results of techno economic analysis are given below (Table 6). For the sake of comparison, an existing Claus plant now in operation at the Minibay GPP was selected (See Figures 30 and 31).

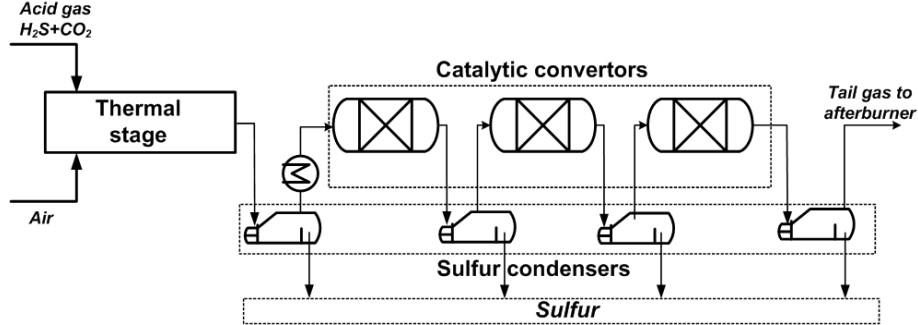

**Figure 30.** Schematic diagram of the Claus plant operating at the Minibay GPP.

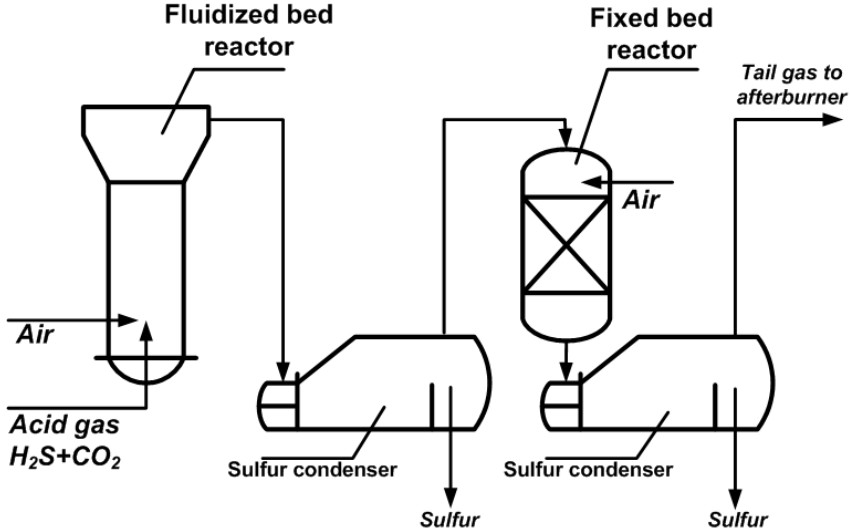

**Figure 31.** Schematic diagram of the alternative direct oxidation plant.

**Table 6.** Comparison of the characteristics of the Claus Pant and a direct oxidation plant with the same capacities [163,164].

| # | Parameters | Direct Oxidation Unit | Three Stage Claus Unit Minibay Gas Processing Plant |
|---|---|---|---|
| 1 | Acid gas ($H_2S+CO_2$) supply, $nm^3/h$ | 1050 | 1050 |
| 2 | $H_2S$ content, %vol. | 80 | 80 |
| 3 | Air supply, $nm^3/h$ | 2000 | 2000 |
| 4 | Sulfur production Annually, ton | 10.000 | 10.000 |
| 5 | Dimensions of the main units | Calculation<br>Fluidized bed reactor:<br>Diameter = 1,5 m<br>Height = 6 m<br>Fixed bed reactor<br>Diameter = 2,5 m<br>Height = 6 m | Direct data<br>Thermal stage furnace<br>Diameter = 2,5 m<br>Length = 7 m<br>Catalytic converters (3 pieces)<br>Diameter = 2,5 m<br>Length = 4 m |
| 6 | Catalyst load, ton | Calculation<br>Fluidized bed reactor-2<br>Fixed bed reactor-5 | Direct data<br>Total: 18 |
| 7 | Sulfur cost Arbitrary units, estimation | 1 | 2.5 |

As shown in the data in Table 6, the installation using the direct oxidation process is significantly more compact, primarily due to the use of a reactor with a fluidized catalyst bed, where the actual target process is effectively combined with the simultaneous removal of excess heat. The required temperature of the direct oxidation process is adjusted by changing the flow rate of the coolant through a heat exchanger placed in the catalyst bed according to fluctuations in the $H_2S$ content in the feed gas. With such technology, capital costs are significantly reduced thanks to the need of fewer parts in the process chain and the significantly lower metal weight. The operational costs are also reduced due to lower energy consumption and the reduced number of required service personnel, which ultimately leads to a decrease in the cost of the final product, i.e., elementary sulfur. The absence of a flame furnace increases the environmental friendliness of the process due to the absence of the formation of toxic side products which occur due to high-temperature interactions of $H_2S$ with $CO_2$-carbonyl sulfide and carbon disulfide.

## 24. Conclusions

An overview of various technologies based on the direct catalytic oxidation of hydrogen sulfide to obtain elementary sulfur is given. Such technologies, primarily their gas-phase version, have obvious advantages, including:

- continuity of the process that allows simultaneous gas purification and the production of a commodity, i.e., elemental sulfur;
- "soft" conditions for implementing the process (T = 220–280 °C) due to the use of a highly active catalyst.

Data on the Claus process and its modern modifications, as the dominant technology for the conversion of hydrogen sulfide into elemental sulfur, are given. The results of research on the development of various catalysts for a direct oxidation process are described. It is shown that catalysts based on transition metal oxides are the most promising.

Oxide catalysts have indisputable advantages over other potential systems, including high thermal stability, low cost of raw materials, and potential for large-scale production, making them optimal in terms of the quality/price ratio, which is a significant indicator for the technical and economic efficiency of commercial processes. This observation is confirmed by the widespread use of Jacobs iron catalysts in SuperClaus installations.

This review also described the results of fundamental studies of the direct catalytic oxidation of hydrogen sulfide, carried out at the Institute of Catalysis SB RAS, on the basis of which industrial installations for hydrogen sulfide removal from gas streams were created.

The industrial facility in the Bavlinskiy gas shop of the PJSC Tatneftegazpererabotka is now in continuous operation.

Several other facilities have been developed and constructed and are now beginning operations:

- An installation for the purification of blow-off gases of high-sulfur crude oil
- An installation for the direct oxidation of hydrogen sulfide as an alternative to the conventional Claus Process
- An installation for the direct oxidation of hydrogen sulfide in the composition of oil-associated gases

The developed technology, in combination with amine treatment, provides:

- The production of commercial products, i.e., fuel gas and sulfur that correspond to technical standards (GOST 5542-87 and GOST 127.1-93, respectively)
- Extended operational range by $H_2S$ content in comparison with Claus units
- Substantial improvement of the environmental situation by avoiding hazardous emissions and the production of waste materials.

**Author Contributions:** Conceptualization, methodology and formal analysis, S.K.; writing, original draft preparation, S.K., M.K., and A.S.; writing, review and editing, S.K., M.K., and A.S.; supervision, Z.R.I. All authors have read and agreed to the published version of the manuscript.

**Funding:** The work was carried out with financial support from the Ministry of Science and Higher Education of RF within the State Assignment for the Boreskov Institute of Catalysis SB RAS (Project No. AAAA-A21-121011390010-7).

**Institutional Review Board Statement:** Not applicable.

**Informed Consent Statement:** Not applicable.

**Conflicts of Interest:** The authors declare no conflict of interest.

**Abbreviations**

| | |
|---|---|
| AC | Activated carbon |
| APG | Associated petroleum gas |
| BAS | Broensted acid sites |
| CNF | Carbon nanofibers |
| CNT | Carbon nanotubes |
| DEA | Diethanolamine |
| EDTA | Ethylenediaminetetraacetic acid |
| DRS | Diffuse reflectance spectra |
| FRC | Federal Research Center |
| FTIR | Furier transform infrared |
| GHSV | Gas hourly space velocity |
| GPP | Gas processing plant |
| JSC | Joint-stock company |
| k | Rate constant |
| LAS | Lewis acid sites |
| LLC | Limited liability company |
| MEA | Monoethanolamine |
| $Nm^3$ | Normal cubic meters |
| OAG | Oil-associated gases |
| ppmv | Part per million by volume |
| PJSC | Public joint-stock company |
| SB RAS | Siberian Branch of the Russian Academy of Sciences |
| W | Reaction rate |
| WHSV | Weight hourly space velocity |
| WHB | Waste heat boiler |

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
