# Peer review of "Direct Selective Oxidation of Hydrogen Sulfide: Laboratory, Pilot and Industrial Tests"

_catalysts, doi:10.3390/catal11091109_

Round 1

Reviewer 1 Report

The paper was revised according to the journal rules. The topic treated deserves to be considered for publication. Low grade fossil fuels are adopted for the elemental sulphur production, several processes are illustrated.

Revisions are required and they are reported below:

  • revise the structure of the paper, english and details ( pag. 21 line 824, centers all figures, )
  • Please add a nomenclature list with all acronyms and variables
  • use ppm by volume or by weight (ppm(v))
  • check and revise figures, e.g. figure 6 "catalist" - y axis
  • numbers the equations 
  • Check and correct the subscripts and superscripts 
  • Citation to be added in pag 22 line 945 
  • Cite fig 11, 12, 13
  •  Instead of "paper" add authors name
  • Fesem and sem post mortem analysis should be added in the paper for solid sorbents 
  • Considerations about sorbents Regeneration could be added  
  • In table 2 and 3 add references 
  • A clear techno economic analysis should be added, there are several case studies showed but they seem incomplete
  • In table 6 the relative order of magnitudine are reported, are compared similar size plants?
  • The pay back period of direct oxidation plant was reported to be 2 y, there are not economical evaluations

Reviewer 2 Report

The authors surveyed the direct oxidative and catalytic processes for H2S conversion into liquid and/or elemental sulfur. It also includes a detailed description of the Claus process as the main reference technology for hydrogen sulfide processing methods. An overview of modern catalytic systems for direct catalytic oxidation technology and known processes is given as well. The review is informative, comprehensive, well-organized, and English is acceptable. The article falls in the scope of catalysts, and I think it can be recommended for publication in catalysts after minor revisions through tacking out the following comments:

  1. The introduction is very short and is not supported by references.
  2. It is good to arrange the activity of dominant nine catalysts based on area, but I would recommend you design a table of more catalysts and arrange them based on stability against temperature as well.
  3. It would be better to mention the rate of oxygen in all methods. This is very critical factor on the oxidation process.
  4. I wonder whether activated carbon for catalytic formation of elemental sulfur is stable at elevated temperature or not. It would be better also to design a table of AC and adsorbents catalysts.
  5. The figures throughout manuscript need significant improvements.
  6. The authors obviously mentioned various materials catalysts for H2S, but it would be better if the authors provide more details about morphology of selected catalysts, for example carbon nanotubes, to reflect the enhancement of the catalytic activity with the chemical engineering.
  7. Since this reaction occurs at elevated temperatures in the presence of oxygen and in some cases, there are byproducts e.g. H2, the authors should write something about safety in this review.
  8. The conclusion should be rewritten again.

Round 2

Reviewer 1 Report

All the suggestions were added to the manuscript